# Recipient tissue microenvironment determines developmental path of intestinal innate lymphoid progenitors

Paula A. Clark [1] ✉, Mayuri Gogoi [1,3], Noe Rodriguez-Rodriguez [1,3], Ana C. F. Ferreira[1], Jane E. Murphy [1], Jennifer A. Walker[1], Alastair Crisp[1], Helen E. Jolin[1], Jacqueline D. Shields [2] & Andrew N. J. McKenzie [1] ✉

Innate lymphoid cells (ILCs) are critical in maintaining tissue homeostasis, and during infection and inflammation. Here we identify, by using combinatorial reporter mice, a rare ILC progenitor (ILCP) population, resident to the small intestinal lamina propria (siLP) in adult mice. Transfer of siLP-ILCP into recipients generates group 1 ILCs (including ILC1 and NK cells), ILC2s and ILC3s within the intestinal microenvironment, but almost exclusively group 1 ILCs in the liver, lung and spleen. Single cell gene expression analysis and high dimensional spectral cytometry analysis of the siLP-ILCPs and ILC progeny indicate that the phenotype of the group 1 ILC progeny is also influenced by the tissue microenvironment. Thus, a local pool of siLP-ILCP can contribute to pan-ILC generation in the intestinal microenvironment but has more restricted potential in other tissues, with a greater propensity than bone marrow-derived ILCPs to favour ILC1 and ILC3 production. Therefore, ILCP potential is influenced by both tissue of origin and the microenvironment during development. This may provide additional flexibility during the tuning of immune reactions.

Innate lymphoid cells (ILC) are commonly located at mucosal surfaces, but are also found in other organs and tissues. Within these sites, tissue-specialised ILC subsets (ILC1, ILC2, ILC3) play key roles in helping to maintain tissue homoeostasis and act as immune sentinels during infection and inflammation[1–3]. Although ILCs are relatively few in number, their capacity to produce high amounts of immunomodulatory cytokines, and ability to interact with stromal, epithelial and other immune cells, makes them potent initiators of immune reactions.

Conventionally, in the adult, ILCs are thought to begin their development in the bone marrow (BM) where an ILC progenitor (ILCP) arises from a common lymphoid progenitor (CLP) before differentiating into more restricted ILC precursors (ILC1P/NKP, ILC2P and ILC3P). Many studies have investigated the transcription factors orchestrating this process and how their coordinated expression determines the fate of progenitors[4–13]. More recently the origins of the developmental trajectories that lead to the discrete ILC subsets have been defined[4,5,14]. Indeed, our previous studies identified a spectrum of multipotent Id2$^+$Bcl11b$^-$ BM populations that were heterogeneous with respect to α4β7 integrin and GATA-3 expression and retained the potential to generate ILC1s, ILC2s, ILC3s and NK cells[5].

However, in addition to these BM-resident ILC progenitors and precursors, there is emerging evidence for ILC progenitors outside of the BM. A population of peripheral human blood CD117$^+$ ILCs which lack mature ILC subset markers, have an ILCP-like transcriptional profile, and give rise to all ILC subsets[15]. Similarly, human foetal liver ILCPs with the potential to produce all 3 ILC sub-types in vitro, and with a similar phenotype (Lin$^-$CD45$^+$CD7$^+$CD127$^+$CD117$^+$) in foetal thymus, intestine, skin, spleen and lung, have also been described[16]. Furthermore, a single cell gene expression atlas of human foetal tissue

[1]MRC Laboratory of Molecular Biology, Cambridge, United Kingdom. [2]Translational Medical Sciences, School of Medicine, University of Nottingham Biodiscovery Institute, Nottingham, United Kingdom. [3]These authors contributed equally: Mayuri Gogoi, Noe Rodriguez-Rodriguez.
✉ e-mail: pclark@mrc-lmb.cam.ac.uk; anm@mrc-lmb.cam.ac.uk

describes a cycling ILCP-like population in the intestine that is positive for *ZBTB16*, *RORC* and *KIT* [17]. Also, within human tonsillar tissue a common Lin⁻CD34⁻CD117⁺ ILCP population produces NK cells, ILC3s and ILC2s, with the two former subsets deviating from the ILC2 lineage[18] – an occurrence mirrored in mouse BM[5]. In addition, Lin⁻CD34⁻CD117⁺CD56⁺ precursors were found to have lost the capacity to generate ILC2s, but retained NK cell and ILC3 potential[18]. Further, a quiescent population of undifferentiated human tonsillar ILCs (Lin⁻CD127⁺HLA-DR⁻NKp44⁻CD62L⁺CD117⁺CD45RA⁺) has the capacity to become activated and differentiate into ILC1s/NKs and ILC3s, but not ILC2s[19]. While Rorγt⁺CD34⁺CD117⁺ ILCPs in human secondary lymphoid tissues give rise to ILC3s[20] or all 3 ILC subsets[21].

Studies in mice have also discovered ILC precursors in extramedullary locations. In the foetal mouse liver primed ILC precursors express markers of commitment such as *Zbtb16, Id2, Rora* and *Itga4*[22]. Foetal mouse gut also contains ILCPs which can differentiate into ILC1s, ILC2s and ILC3s in vitro, express Arginase 1, but are negative for a RORγt fate mapper, ST2, NK1.1 and CD25[23]. Others describe ILCs in the foetal intestine and mesenteric lymph node with gene expression profiles consistent with the ILCPs found in foetal liver and adult BM[24], whilst Simic et al define ILCPs by their expression of PLZF in the periphery of foetal mice[25].

Studies have found PLZF⁺ ILCPs in the adult and neonatal mouse lung[26,27], with a dependence on IGF1 to produce ILC3s in the new born lung[26]. Single cell sequencing data highlighted three clusters with low expression of the mature ILC2 markers *Gata3* and *Il1rl1* (ST2) but increased expression of *Il18r1*, *Thy1*, *Cd7* and *Tcf7*[27], a phenotype similar to the lung ILCPs described by Ghaedi et al.[28] *Il18r1*⁺ST2⁻ cells shared transcriptional similarities with bone marrow ILCPs (BM-ILCPs), and in vitro largely gave rise to ILC2s, but retained the potential to generate small numbers of ILC3s and NK cells/ILC1s[27]. Studies of the adult mouse liver have identified a population of tissue resident foetal-derived lineage negative Sca1⁺Mac1⁺ (LSM) HSCs which preferentially give rise to ILC1s rather than NK cells in the adult liver and expand locally as part of a feedback loop in response to ILC1 produced IFNγ[29].

In light of mounting evidence for extramedullary ILC development from local pools of progenitors and the suggestion that the local microenvironment may influence the fate of these ILCPs we investigate whether the adult mouse small intestinal lamina propria harbour an ILC progenitor population.

To this end, we utilise a multi-coloured fluorescent transcription factor reporter mouse strain to identify a rare ILCP cell population in the gut of adult mouse with the capacity to differentially contribute to ILC tissue subsets dependent upon its ultimate tissue residence. Our findings underline the importance of the tissue microenvironment in directing the development of ILCP to specific ILC subsets.

## Results

### Identification of ILCPs in adult mouse small intestine

The small intestinal lamina propria (siLP) is a rich source of immune cells including ILC1s, ILC2s and ILC3s. However, amongst these differentiated ILC populations, we had observed lineage-negative lymphoid cells that lacked the archetypal markers of mature ILCs or other lymphocytes[5]. We hypothesised that these may include ILCPs. To characterise this population in the adult siLP we generated a multi-allele compound fluorescent reporter mouse to follow the expression of key transcription factors required for ILC development in the siLP (Fig. 1a). These transcription factors were Id2 (BFP reporter) an early determinant of ILC commitment[4,5,12,30], PLZF (Citrine reporter) which is highly expressed in BM-ILCPs[5,6], Bcl11b (tdTomato reporter) and Rorγt (Katushka reporter) key transcription factors in ILC2 and ILC3 development, respectively[31,32].

Using the Id2-BFP reporter with known surface markers we excluded mature siLP-resident ILCs, including CD45⁺lineage⁻Id2-BFP⁺IL-7Rα⁺NK1.1⁺NKp46⁺ ILC1s, CD45⁺lineage⁻Id2-BFP⁺IL-7Rα⁺KLRG1⁺ ILC2s and CD45⁺lineage⁻Id2-BFP⁺IL-7Rα⁺NK1.1⁻NKp46⁺ NCR⁺ILC3s (Fig. 1b). We further excluded CD45⁺lineage⁻Id2-BFP⁺IL-7Rα⁺NK1.1⁻NKp46⁻KLRG1⁻Rorγt-Katushka⁺CCR6⁺/⁻ cells as having already committed to the ILC3 lineage (Fig. 1b). We then subdivided the remaining highly rare CD45⁺lineage⁻Id2-BFP⁺IL-7Rα⁺NK1.1⁻NKp46⁻KLRG1⁻Rorγt-Katushka⁻CCR6⁻ population into four subsets based on their differential expression of PLZF-Citrine and Bcl11b-tdTomato: single positive PLZF⁻Bcl11b⁺ (P⁻B⁺) or PLZF⁺Bcl11b⁻ (P⁺B⁻), double positive PLZF⁺Bcl11b⁺ (P⁺B⁺) and double negative PLZF⁻Bcl11b⁻ (P⁻B⁻) cells (Fig. 1b). We hypothesised that by excluding Bcl11b-tdTomato⁺ committed ILC2s[32] and including PLZF-Citrine positive cells (since PLZF has been used to identify ILCPs in the BM and other extramedullary sites) we would be able to resolve PLZF-Citrine⁺, Bcl11b-tdTomato⁻ (P⁺B⁻) ILC progenitors in the intestine.

To test this hypothesis, we assessed the in vitro expansion potential of the purified P⁺B⁻ cells as compared to the other three PLZF/Bcl11b defined populations (Supplementary Fig. 1). The cells were cultured on OP9 stromal cells (without the expression of Notch signals, but with IL-7 and stem cell factor (SCF)) for three weeks to generate sufficient progeny to permit further analysis. The P⁺B⁻ cell population was considerably more proliferative than the other three sub-populations which showed little-to-no expansion (Fig. 1c). Phenotypic analysis of the progeny from the P⁺B⁻ cells demonstrated that they were dominated by CD45⁺lineage⁻Id2-BFP⁺Rorγt-Katushka⁻CCR6⁻CD4⁻NKp46⁺NK1.1⁺ group 1 ILCs (Fig. 1d and Supplementary Fig. 2). By contrast almost no progeny had marker expression consistent with ILC2s (CD45⁺lineage⁻Id2-BFP⁺Rorγt-Katushka⁻CCR6⁻CD4⁻ICOS⁺Bcl11b⁺) or ILC3s (CD45⁺lineage⁻ID2-BFP⁺Rorγt-Katushka⁺) (Fig. 1d and Supplementary Fig. 2). Furthermore, following stimulation with IL-2, IL-15 and IL-18 the group 1 ILC progeny produced a spectrum of perforin and/or IFN-γ expression confirming their inclusion within the group 1 ILC continuum[33] (Fig. 1e). In vitro analyses using either OP9 or OP9-DL1 stromal cells indicated Notch ligand had no impact on either the proliferative capacity or surface marker phenotype of the progeny (Supplementary Fig. 3).

These results suggest the presence of CD45⁺lineage⁻Id2-BFP⁺IL-7Rα⁺NK1.1⁻NKp46⁻KLRG1⁻Rorγt-Katushka⁻CCR6⁻PLZF⁺Bcl11b⁻ ILCPs located in the adult small intestinal tissue (subsequently referred to as siLP-ILCP), which can give rise primarily to group 1 ILCs in vitro.

### siLP-ILCPs are tissue-resident

Other reports have described the presence of ILCPs in circulation in mouse and human[15,27,34]. To establish if siLP-ILCPs were circulatory or tissue-resident we performed intravascular labelling of circulating CD45⁺ cells[35]. Shortly following intravascular injection of anti-CD45 APC-conjugated antibody, tissues were harvested and analysed for the circulatory labelling of CD45⁺ cells from the blood, spleen, mesenteric lymph node and siLP as compared to all tissue CD45⁺ cells (labelled with anti-CD45 BUV395-conjugated antibody) and cell lineage marker labelling (Fig. 2 and Supplementary Fig. 4). As expected, in the blood almost all lineage-positive cells and the very small population of circulatory ILCs (lineage⁻IL-7Rα⁺) were found to be labelled following exposure to ivCD45 labelling (Fig. 2a,b, Supplementary Fig. 4b). We also detected some ivCD45 labelling of lineage positive cells and ILCs in the spleen presumably due to the exposure of resident lymphocytes to the circulation in the marginal zone[36] (Fig. 2a, b, Supplementary Fig. 4b). By contrast, lineage positive cells and ILCs from the MLN and siLP showed almost undetectable levels of ivCD45 labelling (Fig. 2a, b, Supplementary Fig. 4b), consistent with their largely tissue-resident status[37]. Notably, the siLP-ILCP population was ~100% unlabelled by ivCD45 exposure (Fig. 2a, b). These data establish siLP-ILCPs as a tissue resident population.

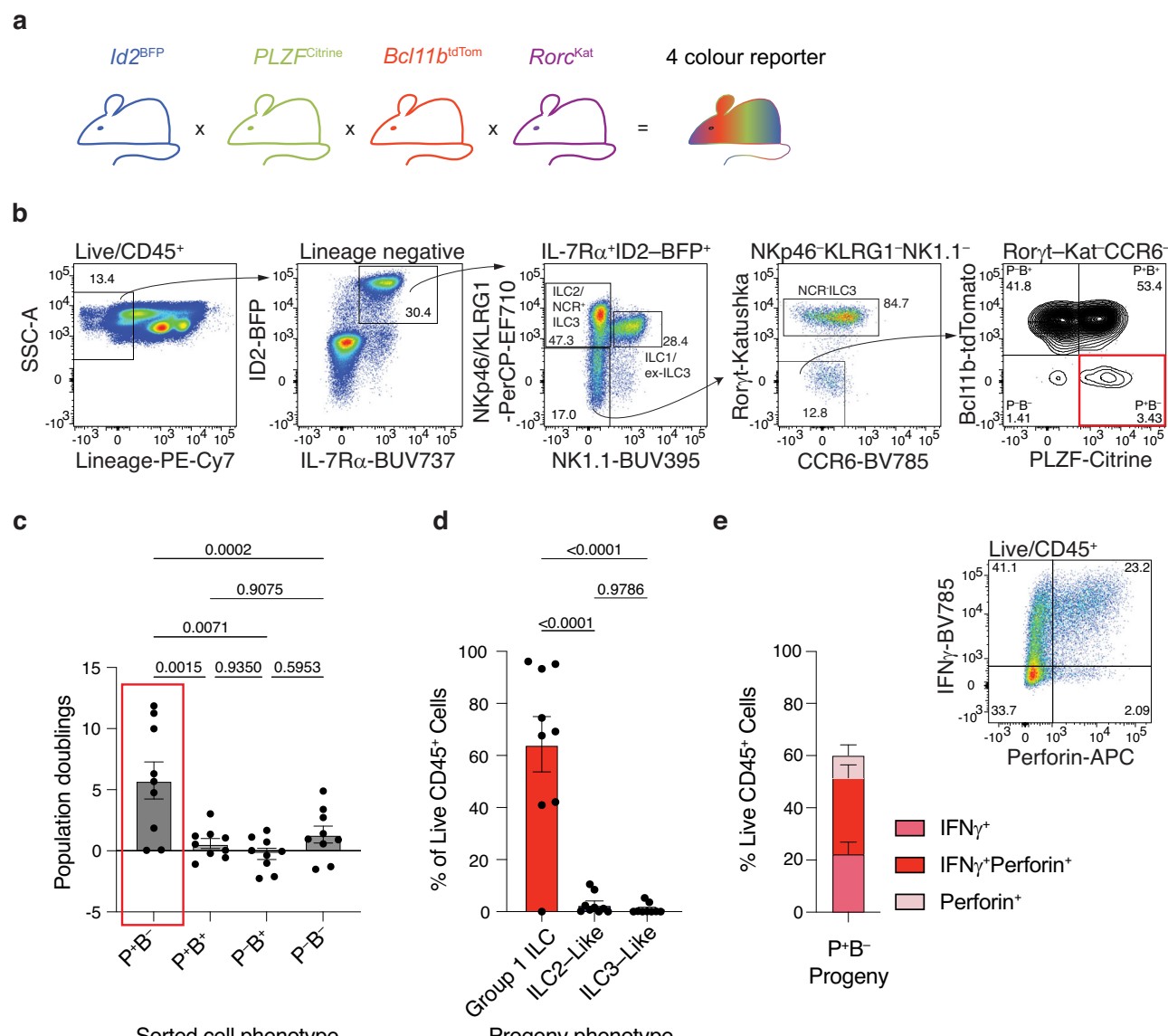

**Fig. 1 | Identification of ILCPs in adult mouse small intestine. a** Schematic representation of the generation of the four-colour reporter mouse (adapted from reference [5]). **b** Definition of siLP ILCs using surface marker and transcription factor fluorescent reporter expression. The ILCs were defined as CD45+lineage-Id2-BFP+IL-7Rα+ (Lineage = CD3, CD4, CD8, CD11b, CD11c, CD19, FcεR1, Ly6G/Ly6C, Ter119). Mature ILC1s, ILC2s and some ILC3s were excluded by the expression of KLRG1, NKp46 and NK1.1. ILC3s were further excluded by the expression of CCR6 and Rorγt-Katushka and the resulting mature marker negative population subdivided into PLZF+Bcl11b- (P+B-), PLZF+Bcl11b+ (P+B+), PLZF-Bcl11b+ (P-B+) and PLZF-Bcl11b- (P-B-) using the PLZF-Citrine and Bcl11b-tdTomato reporters. Numbers associated with boxes indicate the percentage of the parent population. **c** In vitro culture of four populations defined by PLZF and/or Bcl11b under neutral conditions (IL-7 and SCF). FACS sorting strategy shown in Supplementary Fig. 1. Each data point represents the expansion of all the cells of an indicated phenotype from a single mouse. Since the numbers of cells sorted and seeded varies considerably between populations and mice, this is expressed as population doublings. **d** Proportions of ILC progeny produced from PLZF+Bcl11b- (P+B-) cells. Supplementary Fig. 2 defines the phenotype of group 1 ILCs, ILC2-like and ILC3-like progeny. **c,d** Data are cumulative from 2 independent experiments involving a total of 9 mice. Data are plotted as mean with SEM error bars and significance calculated using one-way ANOVA with Tukey's multiple comparisons test. **e** Proportion of IFNγ and perforin positive progeny of P+B- cells. Data are cumulative from 2 independent experiments (7 mice total). The flow cytometry plot shows representative expression of these effectors. Numbers within the quadrants indicate the percentage of the parent population. Data are plotted as mean with SEM error bars. Source data are provided as a Source Data file.

## siLP-ILCP differentially repopulate tissues with group 1 ILCs, ILC2 or ILC3

Next, to investigate the progenitor potential of siLP-ILCPs in vivo we purified CD45.2+ siLP-ILCPs from the 4 colour reporter mice and transferred them intravenously into CD45.1+ *Rag2-/-Il2rg-/-* recipient mice following sublethal irradiation. After a period of 6–7 weeks the liver, lung, spleen and siLP were analysed for the presence and phenotype of donor cell progeny. The siLP-ILCPs were highly proliferative with only a few hundred progenitors, (harvested from multiple donors due to their rarity) leading to thousands of progeny (Supplementary Table 1). Phenotypically the progeny residing in the liver, lung and spleen, like the progeny produced in vitro, were almost exclusively group 1 ILCs (Fig. 3a, Supplementary Fig. 5a, b) with the majority being Eomes positive, although a small proportion of the group 1 ILC progeny were Eomes negative in all tissues examined (Fig. 3b, Supplementary Fig. 5a–d, Supplementary Fig. 6a–c).

Following in vitro stimulation of leucocytes from the liver, lung or spleen, with IL-2, IL-15 and IL-18 the majority of the siLP-ILCP-generated

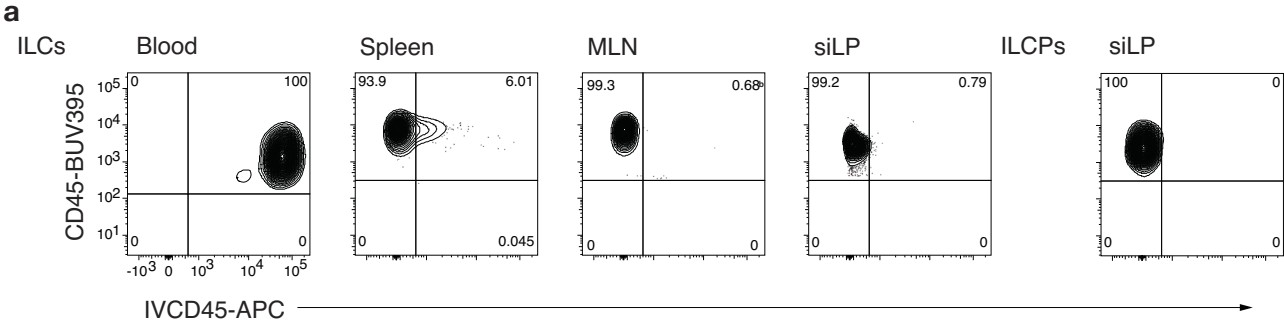

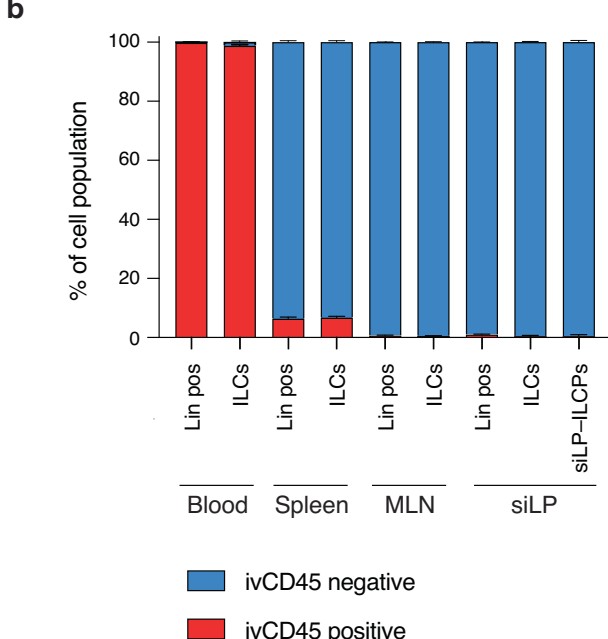

**Fig. 2 | siLP-ILCPs are tissue-resident. a** Representative flow cytometry plots showing ivCD45 labelling of ILCs (LiveCD45⁺lineage⁻IL-7Rα⁺ lineage = CD3, CD4, CD8, CD11b, CD11c, CD19, FcεR1, Ly6G/Ly6C, Ter119) in the four tissues examined and ILCPs in the siLP only (LiveCD45⁺lineage⁻IL-7Rα⁺NKp46⁻NK1.1⁻KLRG1⁻CCR6⁻Rorγt-Katushka⁻PLZF⁺Bcl11b⁻). Numbers within the quadrants are the percentage of the parent gate. Gating strategy shown in Supplementary Fig. 4a. **b** Proportions of each population from each tissue which are ivCD45 positive or negative. Data are from 6 mice from one experiment and are representative of 3 independent experiments and plotted as mean with SEM error bars. Source data are provided as a Source Data file.

group 1 ILC progeny produced the effector molecules perforin and/or IFN-γ (Fig. 3c, d), though the proportion of cells making perforin and/or IFN-γ varied between these tissues (Fig. 3c, d). We further assayed the cytotoxic activity of the progeny isolated from the livers of the mice which received siLP-ILCP (Fig. 3e). Purified siLP-ILCP-derived progeny, purified mature NK cells (from wildtype spleen) or CD4⁺ T cells (as negative controls also from wildtype spleen) were co-cultured with B16F10 melanoma cells and their killing assessed. As expected, NK cells showed cytotoxic activity towards the melanoma cells, whilst the CD4⁺ T cells did not (Fig. 3e). Notably, the progeny from the livers of siLP-ILCP recipient mice efficiently killed the B16F10 target cells, confirming that they have cytotoxic activity (Fig. 3e). Thus, siLP-ILCP-derived group 1 ILCs have a phenotype and the functional properties characteristic of the group 1 ILC continuum.

Surprisingly, however, transferred siLP-ILCPs gave rise to group 1 ILCs, ILC2s and ILC3s in the siLP microenvironment (Fig. 4a–c). This characterisation was confirmed by analysis for the archetypal NK cell and ILC2 transcription factors Eomes and GATA-3, respectively (Supplementary Fig. 6a–c). Within the Rorγt-Katushka⁺ ILC3 population there were both IL-7Rα positive and negative cells. ILC3s have been reported to down-regulate IL-7Rα when stimulated[38], and IL-7 engagement can lead to down-regulation of its own receptor, which may explain this observation especially in the context of irradiated mice in which there is no competition for the IL-7 produced by the radio-resistant cells[39]. The Rorγt-Katushka⁺ progeny also included both NKp46⁺ and NKp46⁻ cells indicative of the presence of NCR-positive and negative ILC3s identified previously[40] (Fig. 4a–c). Moreover, some expressed NK1.1 indicating this population may also include ex-ILC3s in transition to a group 1 phenotype (Fig. 4a)[41]. The proportions of these progeny populations in the siLP were significantly different from the other tissues with the exception of the NCR⁻ILC3-like cells which likely did not reach statistical significance as they were present in very small numbers and not found in all recipients (Fig. 4c). It has been reported that BM-ILCPs retain the potential to produce all 3 ILC sub-types across multiple tissues in *Rag2⁻/⁻Il2rg⁻/⁻* recipients[4,28,33,34]. Thus, the lineage restriction of the siLP-ILCPs in the lungs of *Rag2⁻/⁻Il2rg⁻/⁻* mice appeared to contrast with these reports. We therefore transferred either BM-ILCPs (sorted as CD45⁺lineage⁻ID2-BFP⁺IL7Rα⁺Flt3⁻Sca1⁻CD25⁻Bcl11b-tdTomato⁻Rorγt-Katushka⁻PLZF^hiα4β7^hi), or siLP-ILCPs, into *Rag2⁻/⁻Il2rg⁻/⁻* recipients to directly compare the phenotype of their progeny

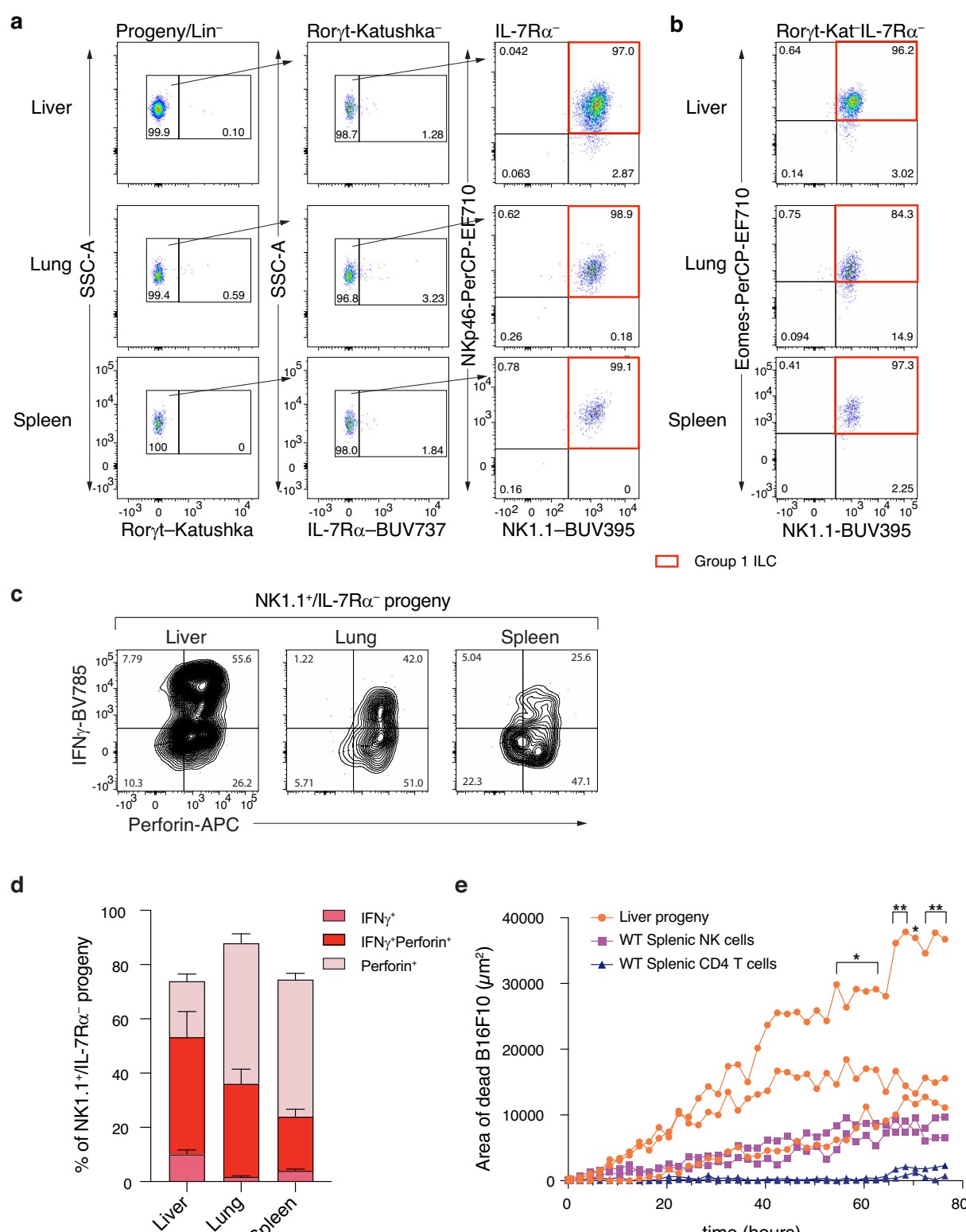

in the lung and siLP. We found that the BM-ILCPs gave rise to significantly more ILC2-like cells in the lung and siLP of recipients than the siLP-ILCPs which generated very few ILC2-like cells in the lung. By contrast the siLP-ILCPs produced many more ILC3-like cells in the intestine than were generated by the BM-ILCPs (Fig. 4d).

Together, these results demonstrate that siLP-ILCPs are multipotent but the identity of their progeny is significantly influenced by the microenvironmental cues. Moreover, ILC subtype developmental potential is also impacted by the ILCP tissue of origin.

## Single cell gene expression analysis confirms multipotency of siLP-ILCPs and the influence of microenvironment on their fate
To better define the siLP-ILCP progenitor population and its progeny in vivo, particularly in the siLP, we utilised single cell RNA sequencing

**Fig. 3 | siLP-ILCPs repopulate liver, lung and spleen with group 1 ILCs.**
**a** Representative flow cytometry plots defining the cell surface and reporter phenotype of donor progeny in the three tissues indicated. (Lineage = CD3, CD4, CD8, CD19, Ter119). Numbers within plots are the percentage of the parent gate. Gating strategy to define lineage negative progeny is shown in Supplementary Fig. 5a and definition of NKp46 positive and negative populations in Supplementary Fig. 5b. **b** Representative flow cytometry plots defining the phenotype of donor progeny in the three tissues indicated with respect to expression of the transcription factor Eomes. Numbers within quadrants are the percentage of the parent gate. Gating strategy for Rorγt⁻Kat⁻IL7Rα⁻ as for (**a**). Definition of positive and negative populations for NK1.1 and Eomes is shown in Supplementary Fig. 5c,d. **c** Representative flow cytometry plots of the expression of perforin and IFNγ by NK1.1⁺IL-7Rα⁻ progeny from the indicated recipient mouse tissues, following 24-hour stimulation with IL-2/IL-15/IL-18. Gating strategy for donor progeny is shown in Supplementary Fig. 5a. Numbers in quadrants are percentages of parent gate. **d** Quantification of

perforin and IFNγ production by progeny. Data are cumulative from 8 recipient mice analysed over 3 independent experiments and plotted as mean with SEM error bars. **e** In vitro cytotoxic activity of progeny isolated from recipient livers. Cytotoxic activity was measured as area of dead B16F10 cells and compared to the activity of wildtype (WT) splenic NK cells (positive control, magenta lines) and CD4⁺ T cells (negative control, blue lines). Orange lines show the activity of the siLP-ILCP progeny from the liver from 3 individual mice analysed in one experiment and are representative of 2 independent experiments. *p-values are as follows: 54.4hrs = 0.0383, 56.4 h = 0.0403, 58.4 h = 0.0291, 60.4 h = 0.0148, 62.4 h = 0.0235, 70.4 h = 0.0122, **p-values are as follows: 66.4 h = 0.0072, 68.4 h = 0.0064, 72.4 h = 0.0079, 74.4 h = 0.0054, 76.4 h = 0.0085, when comparing the average killing activity of the liver progeny cells from the 3 mice to the average of the negative control CD4⁺ T cells (2-way ANOVA with Šídák's multiple comparisons test). Source data are provided as a Source Data file.

(scRNA-seq) analysis. We purified the siLP-ILCPs from the 4-colour reporter-mouse, and their CD45.2⁺Id2-BFP⁺ progeny from the siLP and lungs of *Rag2⁻/⁻Il2rg⁻/⁻* (CD45.1⁺) recipient mice. For comparison, we included data from lung group 1 ILC progeny produced by BM-derived aceNKPs (lineage⁻Id2⁺IL-7Rα⁺CD25⁻α4β7⁻NKG2A/C/E⁺Bcl11b⁻ progenitors)[33]. We identified 5 clusters on the UMAP plot, with the siLP-ILCP being distinct from their progeny, both from the lung and the siLP (Fig. 5a). Moreover, the siLP progeny clearly fell into 3 distinct mature ILC clusters (C1, C2 and C3) with characteristics of group 1 ILCs, ILC2s and ILC3s, respectively. By contrast, in the lung microenvironment only group 1 ILCs were identified, confirming our assignment of these progeny using flow cytometry. The group 1 cluster (C1) included the expression of killer-like receptor genes (e.g. *Klrb1c, Klra7, Klrd1, Klrc2, Klrk1*), *Eomes* (encoding Eomesodermin) and *Gzma* (encoding Granzyme A) (Fig. 5b–d). The ILC2 cluster (C2) included genes encoding the type-2 cytokines IL-13, IL-4 and IL-5, and arginase 1 and IL-17RB (IL-25 receptor) (Fig. 5b–d). The ILC3 cluster (C3) was distinguished by the expression of transcripts encoding IL-22 and the IL-23 receptor (Fig. 5b–d). Furthermore, mirroring the flow cytometry data *Il7r* gene expression was only detected in a subset of ILC3s (Fig. 5d). We additionally used ILC subset gene signatures, reported previously[42,43], to place our assignment of the siLP-ILCP progeny in context. We found that a number of the genes that were most differentially expressed between our progeny clusters were consistent with these signatures (Fig. 5b, red; group 1 ILCs, blue; ILC2 and green; ILC3). Additionally, we found good concordance between the gene signatures which define group 1, 2 and 3 ILCs[42,43] and those genes which cluster populations C1 – C3 (Supplementary Fig. 7 and Supplementary Data 1).

The gene expression profiles of the lung emergent progeny from siLP-ILCPs and those arising from aceNKPs indicated that these group 1 ILCs were indistinguishable (Fig. 5a). However, comparison of the gene expression profiles of the siLP-ILCP-derived group 1 ILC progeny found in the siLP with those located in the lungs of the recipients indicated that they formed distinct clusters (Fig. 5a). This separation may be partially explained by the differential expression of genes which demarcate 'tissue-specific' (*Kit, Asb2, Cxcr6* and *Emb*) or 'non-tissue-specific' (*Cma1, S100a4, Itgam* and *Zeb2*) group 1 ILCs, as proposed by McFarland and colleagues[44]. In this context the group 1 ILCs arising in the siLP appear more 'tissue-specific' whilst those arising in the lung are more 'non-tissue-specific' which could also be consistent with a circulating NK cell phenotype (Supplementary Fig. 8a,b). Further, we observed differentially higher expression of *Il21r* in the gut-associated group 1 ILC progeny (a gene whose transcription has been associated with tissue-resident ILC1s in the salivary gland and the liver (Supplementary Fig. 8a and b)[45,46]. We also analysed the scRNAseq data from the group 1 progeny from the lung and siLP using markers such as CD49a (Itga1) and CD49b (Itga2) and Ly49 markers (Klra3, 4, 5, 7, 8 and 9) in an attempt to assign them to an ILC1 or NK cell identity. The group 1 ILC lung cluster showed a greater proportion of cells expressing

CD49b, Klra3 and Klra9. However, both clusters express CD49a, Klra5 and Klra7 (Supplementary Fig. 8c), which precluded partitioning of the progeny into the two classically proposed cell types. By analysing the scRNAseq data both as an average across the group 1 ILC clusters (Fig. 6a) and on a cell-by-cell basis (Fig. 6b) using gene expression signatures that have been proposed to be informative in defining ILC1s and NK cells[42,44] we were unable to discriminate two non-overlapping populations. Recent studies have identified NK cells in tissues and tumours which express tissue residency markers in response to immunological challenge which parallel some of the complexity of group 1 progeny phenotypes we observe[47–49]. Whilst the expression profiles of the siLP progeny could reflect a mixture of tissue-resident NK cells and ILC1s, overall, in the absence of a clear dichotomy along an NK cell/ILC1 axis for all the group 1 progeny our observations are more consistent with them lying on a continuum of group 1 ILCs.

## Tissue microenvironment differentially alters siLP-ILCP-derived group 1 ILC phenotypes

To better understand the influence of the tissue microenvironment on the phenotype of the siLP-ILCP progeny we applied multi-parameter spectral cytometry to phenotype group 1 ILC progeny (CD45.2⁺Id2-BFP⁺NK1.1⁺) in the BM, lung, liver, spleen and siLP of *Rag2⁻/⁻Il2rg⁻/⁻* (CD45.1⁺) recipient mice which received siLP-ILCPs. Cluster analysis identified 8 clusters (LP1 – LP8) (Fig. 7a). Clusters LP1 and LP2 have NK cell-like features, expressing the highest levels of Eomes and CD49b and the lowest levels of Granzyme C and CXCR6, and lower levels of CD49a (Fig. 7b and Supplementary Fig. 9a)[50,51]. The other clusters, with the exception of LP5 which appears to be a very small outlying cluster, have phenotypes more consistent with ILC1s having higher levels of CD49a, Granzyme C, CD200R and CXCR6, and lower levels of CD49b and Eomes (Fig. 7b)[50,51]. Of these LP6 and LP7 are notable for having high levels of Bcl11b-tdTomato and the lowest levels of CD49b (Fig. 7b). However, despite these clusters having recognisable features of NK cells and/or ILC1s, we could not discern evidence for an ILC1/NK cell dichotomy. We also examined the distribution of these progeny clusters across tissue sources (Fig. 7c, d, Supplementary Fig. 9b) and observed that the liver had the broadest mix of clusters whilst in the BM, lung and spleen the predominant contributor was cluster LP2 (Fig. 7c, d, Supplementary Fig. 9b). By contrast although cluster LP2 contributed more than 50% of the siLP group 1 ILC progeny there was considerable contribution from the other more ILC1-like clusters (Fig. 7c,d, Supplementary Fig. 9b).

These results confirm that a majority of the lung progeny have a more NK cell-like phenotype and that the siLP progeny are more mixed with a larger contribution from the more ILC1-like clusters. However, as observed by scRNAseq analysis, the group 1 ILC progeny are better described as having a spectrum of phenotypes which varies dependent on the local tissue which they colonise.

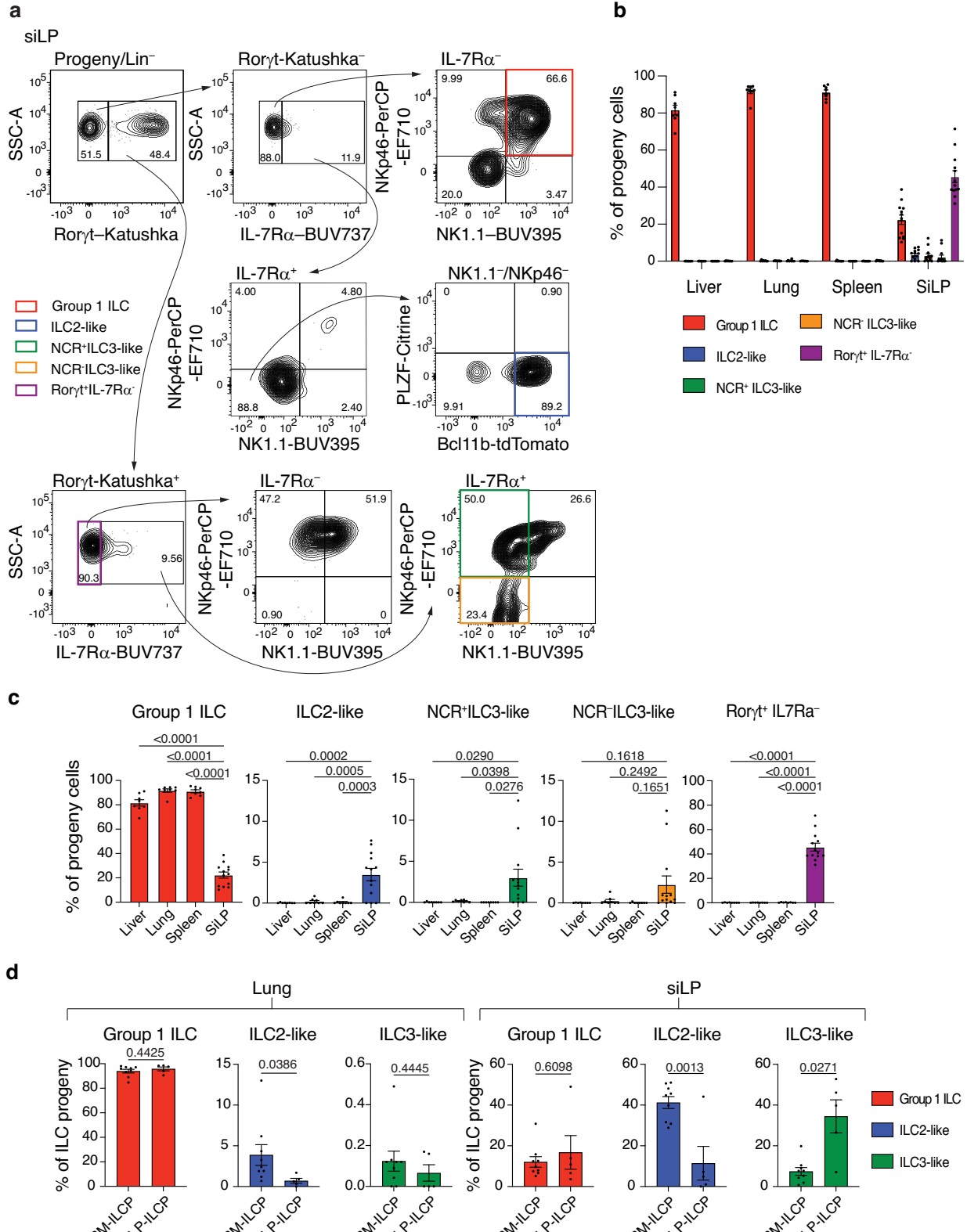

## Single cell gene expression analysis shows commonalities between siLP-ILCPs and BM-ILCPs

In our initial analysis of the gene expression profiles of both the siLP-ILCP and their progeny in the gut we observed that the progenitor cluster was distinct from the progeny (Fig. 5a). Moreover, when the gene expression profile of siLP-ILCP was compared on a cell-by-cell basis with the signatures for their progeny we confirmed that the siLP-

ILCPs did not contain populations that were already committed to an ILC subset, having expression of genes consistent with all subsets (Fig. 5c). A similar picture emerged when we interrogated the siLP-ILCPs gene expression data with published ILC gene signatures[42,43], which showed that siLP-ILCPs were enriched for genes characteristic of all 3 subsets (Supplementary Fig. 7). The scRNAseq data also confirmed that siLP-ILCPs, in addition to being negative for the characteristic ILC3

**Fig. 4 | siLP-ILCPs generate group 1 ILCs, ILC2s and ILC3s in the small intestine.** **a** Representative flow cytometry plots defining the cell surface and reporter phenotype of donor progeny in the siLP of recipients. (Lineage = CD3, CD4, CD8, CD19, Ter119). Numbers within gates are the percentage of the parent gate. The gating strategy to define lineage-negative progeny is shown in Supplementary Fig. 5a. **b** Quantification of the proportions of progeny in each tissue of each phenotype defined by surface markers and reporters. Group 1 ILCs are defined as CD45.2⁺Id2-BFP⁺lineage⁻Rorγt-Katushka⁻IL-7Rα⁻NKp46⁺NK1.1⁺ (red), ILC2-like are defined as CD45.2⁺Id2-BFP⁺lineage⁻Rorγt-Katushka⁻IL-7Rα⁺NKp46⁻NK1.1⁻Bcl11bʰⁱPLZF⁻ (blue), NCR⁺ILC3-like are defined as CD45.2⁺Id2-BFP⁺lineage⁻Rorγt-Katushka⁺IL-7Rα⁺NKp46⁺NK1.1⁻ (green), NCR⁻ILC3-like are defined as CD45.2⁺Id2-BFP⁺lineage⁻Rorγt-Katushka⁺IL-7Rα⁺NKp46⁻NK1.1⁻ (yellow) with an additional population defined as CD45.2⁺Id2-BFP⁺lineage⁻Rorγt-Katushka⁺IL-7Rα⁻ (purple). Data are cumulative from 5 independent experiments involving a total of 8 (lung,

liver, spleen) or 13 (siLP) recipient mice and plotted as mean with SEM error bars. **c** Analysis of the statistical significance between the proportions of the progeny populations in the siLP compared with those in the liver, lung and spleen (one-way ANOVA with Tukey's multiple comparisons test). Data are cumulative from 5 independent experiments involving a total of 8 (lung, liver, spleen) or 13 (siLP) mice and plotted as mean with SEM error bars. **d** Comparison of the in vivo progeny phenotypes of BM-ILCPs and siLP-ILCPs in the lung and siLP of recipients. ILC progeny defined as Live CD45.2⁺Id2-BFP⁺ lineage negative. Group 1 ILC defined as NK1.1⁺Rorγt⁻, ILC2-like defined as NK1.1⁻Rorγt⁻Bcl11b-tdTomato⁺PLZF-Citrine⁻, ILC3-like NK1.1⁻Rorγt⁺. Data is pooled from 2 independent experiments where *n* = 9 recipients of BM-ILCPs (male and female of 8 to 22 weeks of age) and 5 recipients of siLP-ILCPs (male and female of 11 to 22 weeks of age). Data is plotted as mean with SEM error bars and significance calculated by a two-way unpaired t-test with Welch's correction when required. Source data are provided as a Source Data file.

transcription factor reporter Rorγt-Katushka, were also negative for the mature group 1 ILC transcription factor *Tbx-21* (T-bet) and low for *Gata-3*, the high expression of which is characteristic of ILC2s (Supplementary Fig. 10).

We next compared the gene expression profile of the siLP-ILCPs with that for the the siLP-ILCP-derived progeny arising in the siLP, to determine the relative expression patterns of genes associated with ILC progenitors in the BM. We found that *Zbtb16* (PLZF), *Pdcd1* (PD-1), *Sox4*, *Tcf7* (TCF-1) and *Il7r* (IL-7Rα), all genes that have been reported to be expressed by BM-ILCPs[9,13,30,52,53], were expressed at a higher level than in all or most of the progeny from the siLP (Fig. 8a). Integrin α4β7 is also a known marker of BM-ILCPs[13] and we determined that *Itga4* encoding the integrin α4 chain was more highly represented in the siLP-ILCPs than in their progeny, whilst *Itgb7* (encoding integrin β7 chain) was expressed in all populations (Fig. 8a). *Kit*, found on HSCs, CLPs and ILCPs, was expressed by both progenitors and progeny, whereas *Flt3* and *Ly6a* (Sca1), markers of CLPs, were not expressed by the progenitors (Fig. 8a). These data suggested that siLP-ILCPs share aspects of their transcriptional profile with BM-ILCPs[5]. ILCPs from foetal mouse intestine with the potential to become ILC1s, ILC2s and ILC3s in vitro have been described previously to be marked by Arginase1[23]. We found that only a proportion of siLP-ILCPs from adult siLP expressed *Arg1* (Fig. 8a) or its protein product (Supplementary Fig. 11a) and were not defined by its expression. A lung-resident ILCP has also been described which expresses *Il18r1*[27,28] but although some siLP-ILCPs expressed *Il18r1* and its protein product, they were not defined by this marker (Fig. 8a, Supplementary Fig. 11b). Also noteworthy was the expression of *Cd7* by siLP-ILCP which appears in BM precursors exhibiting a more ILC1/NK/ILC3 trajectory[5,33]. In an unbiased assessment of those genes whose relative expression was higher in the progenitors than each of their progeny subsets we identified a signature that distinguished the progenitors (Fig. 8b, c).

### CD45⁺lineage⁻Id2-BFP⁺IL-7Rα⁺NK1.1⁻NKp46⁻KLRG1⁻Rorγt-Katushka⁻CCR6⁻PLZF⁺Bcl11b⁻ cells can be found in multiple tissues

Having defined a very rare population of ILCPs in the adult mouse siLP we used our reporter combination and gating strategy to determine if CD45⁺lineage⁻Id2-BFP⁺IL-7Rα⁺NK1.1⁻NKp46⁻KLRG1⁻Rorγt-Katushka⁻CCR6⁻PLZF⁺Bcl11b⁻ cells were present in other tissues. We found a comparable population in the BM, spleen, lung, liver and mesenteric lymph node (Supplementary Fig. 12a,b). Approximate average yield of these cells per organ, where the entire organ was processed (siLP, spleen, lung, liver and mesenteric lymph node), is shown in Supplementary Fig. 12c. By contrast, we did not find this population in the colonic lamina propria, amongst small intestinal intra-epithelial lymphocytes or in the fat (Supplementary Fig. 12b). We then combined the spectral cytometric data for this population from all the tissues and performed cluster analysis, identifying 8 clusters

(ILCP1 - ILCP8) (Fig. 9a). The contribution of each cluster to the population in each of the tissues varied subtly from site to site. All tissues had contributions from multiple clusters, but the BM had a major contribution from cluster ILCP3 which shows higher expression of α4β7 and PD-1, consistent with published descriptions of BM-ILCPs[6,9,13,30,53] (Fig. 9b–e, Supplementary Fig. 12d). The lung had a significant contribution from ILCP1 and almost 90% came from clusters expressing the highest levels of IL18Rα (ILCP1-ILCP5), consistent with the phenotype of lung ILCPs described previously[27,28] (Fig. 9b–e). Notably cluster ILCP7 was quite distinct and mostly restricted to the liver (Fig. 9a, c and d). This cluster had higher levels of CXCR6, CD49a and Thy1, and expressed CD122, all of which are expressed by the liver ILC1 precursors described by Bai et al.[29] (Fig. 9b). Cell-by-cell analysis for a restricted set of markers (Fig. 9f) confirmed that some, but not all, siLP-ILCPs express IL-18Rα and Arginase 1 and that many but not all express CD122, c-Kit, α4β7 and PD-1 (Fig. 9f). Overall, comparison of our high dimensional spectral cytometric data with previous descriptions of murine extramedullary ILCPs[22–25,27–29] indicates similarities with those described in studies of other tissues.

Taken together, our data support the existence of a population of multipotent ILCPs which are resident in the intestine of adult mice and capable of giving rise to group 1 ILCs, ILC2 and ILC3 in the siLP microenvironment. We also found evidence that this population exists at other peripheral sites. Interestingly, the potential of our siLP-ILCPs outside their tissue microenvironment of origin is significantly restricted, with group 1 ILCs being almost exclusively emergent. Moreover, the lineage potential of these siLP-ILCPs is heavily skewed in favour of ILC3, and away from ILC2s, as compared with progeny from BM-ILCPs. This demonstrates the importance not only of the tissue microenvironment for ILCP maturation but also their tissue of origin.

## Discussion

The small intestinal lamina propria of adult mice is an immune cell-rich tissue harbouring a diversity of leucocytes with cell surface marker expression typical of generic lineage commitment, but also complex molecular repertoires strongly influenced by tissue-restricted micro-environmental cues. By creating a mouse strain carrying a combination of transcription factor reporters we have been able to identify a previously unappreciated intestine-resident ILC progenitor population amongst the variety of lymphocytes within the siLP of adult mice. PLZF (encoded by *Zbtb16*) has been used as a defining marker of ILCPs in many studies[4–7,15,26,27], but the combination of this with the other transcription factor reporters allowed the identification of highly infrequent ILCPs in adult gut tissue. Single-cell gene expression analysis indicated that these siLP-ILCPs could not be subdivided into further discrete subclusters and did not include more developmentally mature ILC subsets. Indeed, cells within the siLP-ILCP population were characterised by a spectrum of gene expression profiles consistent with the observed heterogeneity of haematopoietic progenitor cells[54].

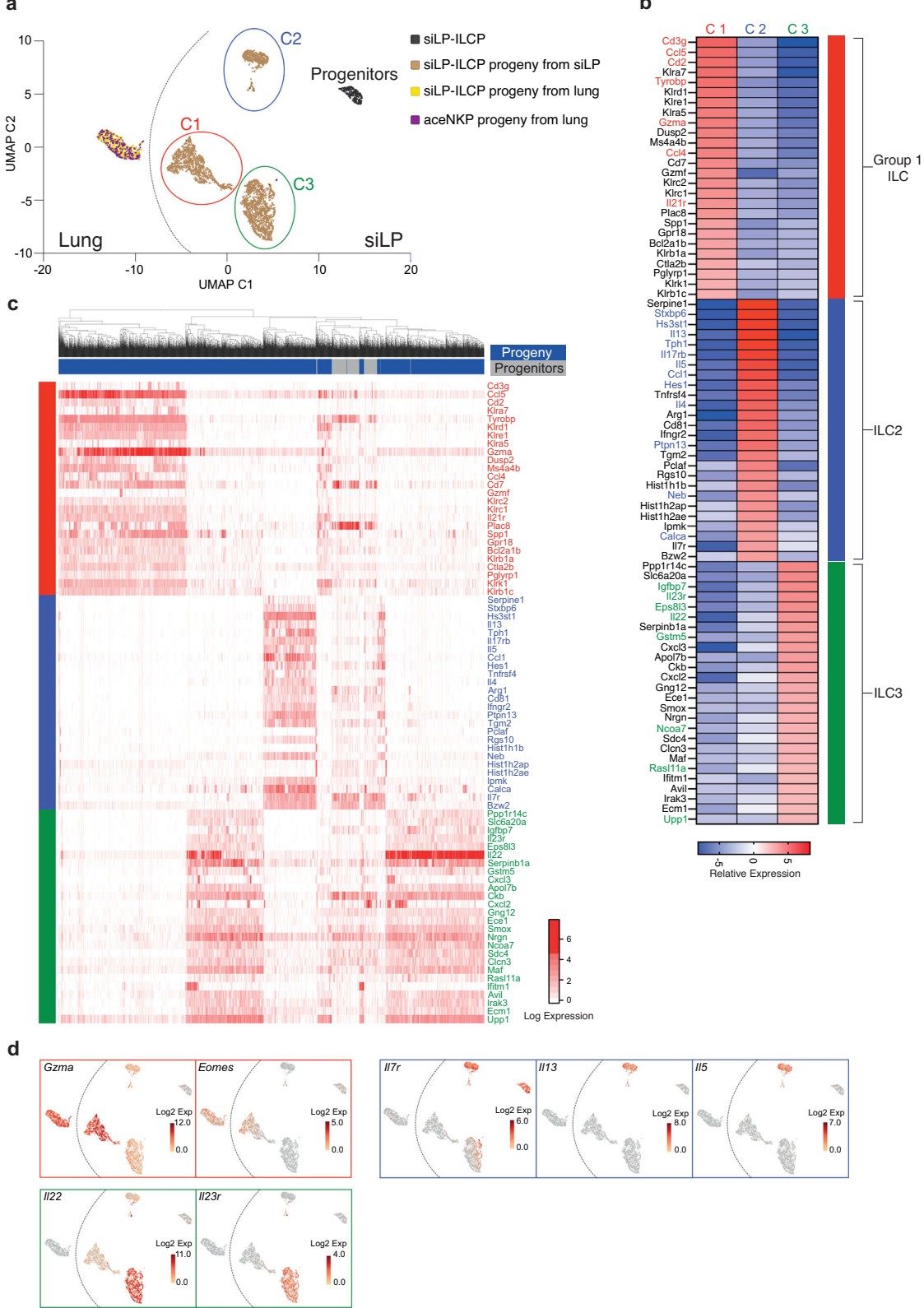

**Fig. 5 | scRNA-seq analysis identifies discrete siLP-ILCPs and ILC progeny.**
**a** UMAP plot of single cell gene expression analysis of siLP-ILCPs, and siLP and lung progeny (4399 individual cells) purified from sublethally irradiated $Rag2^{-/-} Il2rg^{-/-}$ recipients of adoptive transfer of siLP-ILCPs. A small number (334 cells) of lung progeny from aceNKP adoptive transfer was also sampled for comparison[33] (GSE213814). **b** Heatmap of the top 25 genes differentially expressed in siLP progeny clusters defined in (**a**). Previously reported signature genes of group 1, 2 and 3 ILCs are highlighted in red, blue and green respectively. **c** Heatmap of the expression of the top 25 genes differentially expressed by the siLP progeny and the siLP-ILCP progenitors. Log expression is shown normalised by column. Columns represent 374 individual siLP-ILCP cells (highlighted in grey) and 3425 siLP progeny cells from cluster 1, 2 and 3 (highlighted in blue) rows represent the different genes. **d** UMAP plot with expression level (log2 expression) of key differentially expressed genes per individual cell.

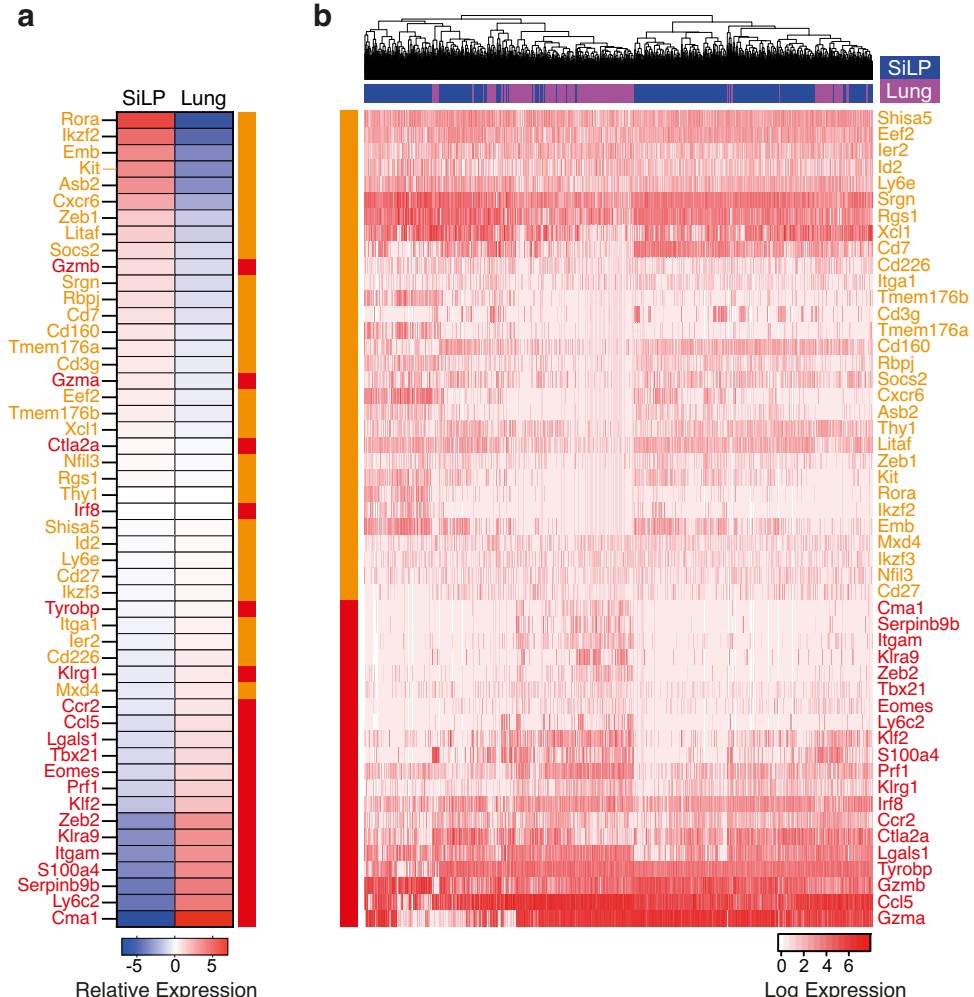

**Fig. 6 | scRNAseq analysis places siLP-ILCP group 1 ILC progeny in the lung and siLP on a phenotypic continuum. a** Heatmap of the relative expression of published NK cell (highlighted in red) and ILC1 (highlighted in orange) signature genes of the siLP and lung group 1 ILC progeny clusters of the siLP-ILCPs. **b** Heatmap of the expression of published NK cell (highlighted in red) and ILC1 (highlighted in orange) signature genes of the siLP and lung group 1 ILC progeny. Log expression is shown normalised by column. Columns represent 1247 individual siLP group 1 ILC progeny (highlighted in blue) and 673 lung group 1 ILC progeny (highlighted in purple) rows represent the different genes.

Our multi-parametric analysis of protein marker expression of the siLP-ILCPs and cells with the same surface marker and reporter profile from multiple tissues also highlighted a diversity of phenotypes across and within tissues. Comparison of these phenotypes with those reported in the literature for ILCPs found in mouse foetal liver, foetal peripheral tissues and adult peripheral tissues[22–25,27–29] identified commonalities but not absolute concordance. For example, arginase 1 is reported to mark in vitro validated ILCPs from the foetal mouse gut and a phenotypically similar population in the adult[23] whilst only a subset of siLP-ILCPs expressed this marker. In addition, whilst in the lung we saw an enrichment for those ILCP clusters that had the highest IL18Rα expression consistent with previous studies[27,28], in the siLP only a subset of the ILCPs expressed this receptor. Further, whilst our ILCP7 cluster, found almost exclusively in liver, shared the markers CXCR6, CD49a, CD122 with the adult liver ILC1Ps described by Bai et al. [29], this was not the only cluster that we found in the liver.

We used gene expression data to generate a signature that distinguished siLP-ILCPs from their intestinal progeny. However, the diversity of expression profiles observed for protein markers within and across tissues makes the extraction of a more detailed common signature challenging even within the mouse and even more so when compared to human. Moreover, studies of ILCPs in human secondary lymphoid tissues and pre-natal liver[16,21] suggest they express Rorγt, which is a notable difference from mouse siLP-ILCPs which are all Rorγt-Katushka reporter negative. The tissue residence of siLP-ILCPs also differs from reported human peripheral blood ILCPs which act as a reservoir for naïve ILCs that can populate tissues and differentiate appropriately to the immune challenge[15,19]. Indeed, the siLP-ILCPs are completely negative for intravascular CD45 labelling. This also sets them apart from previously described mouse lung ILCs, 30-40% of which were labelled by ivCD45 and could therefore represent cells transiting through the pulmonary circulation[27]. This observation argues against the recent recruitment of siLP-ILCPs from the circulation and is more consistent with their persisting from foetal progenitors seeded from the foetal liver or BM during the neonatal period, or very slow replenishment from the adult BM[55].

Our initial in vitro assessment of the potential of the CD45+lineage−Id2-BFP+IL-7Rα+NK1.1−NKp46−KLRG1−Rorγt-Katushka−CCR6−PLZF+Bcl11b− cells confirmed our hypothesis that they might have progenitor-like properties but suggested their potential was limited to group 1 ILCs. However, when siLP-ILCPs were

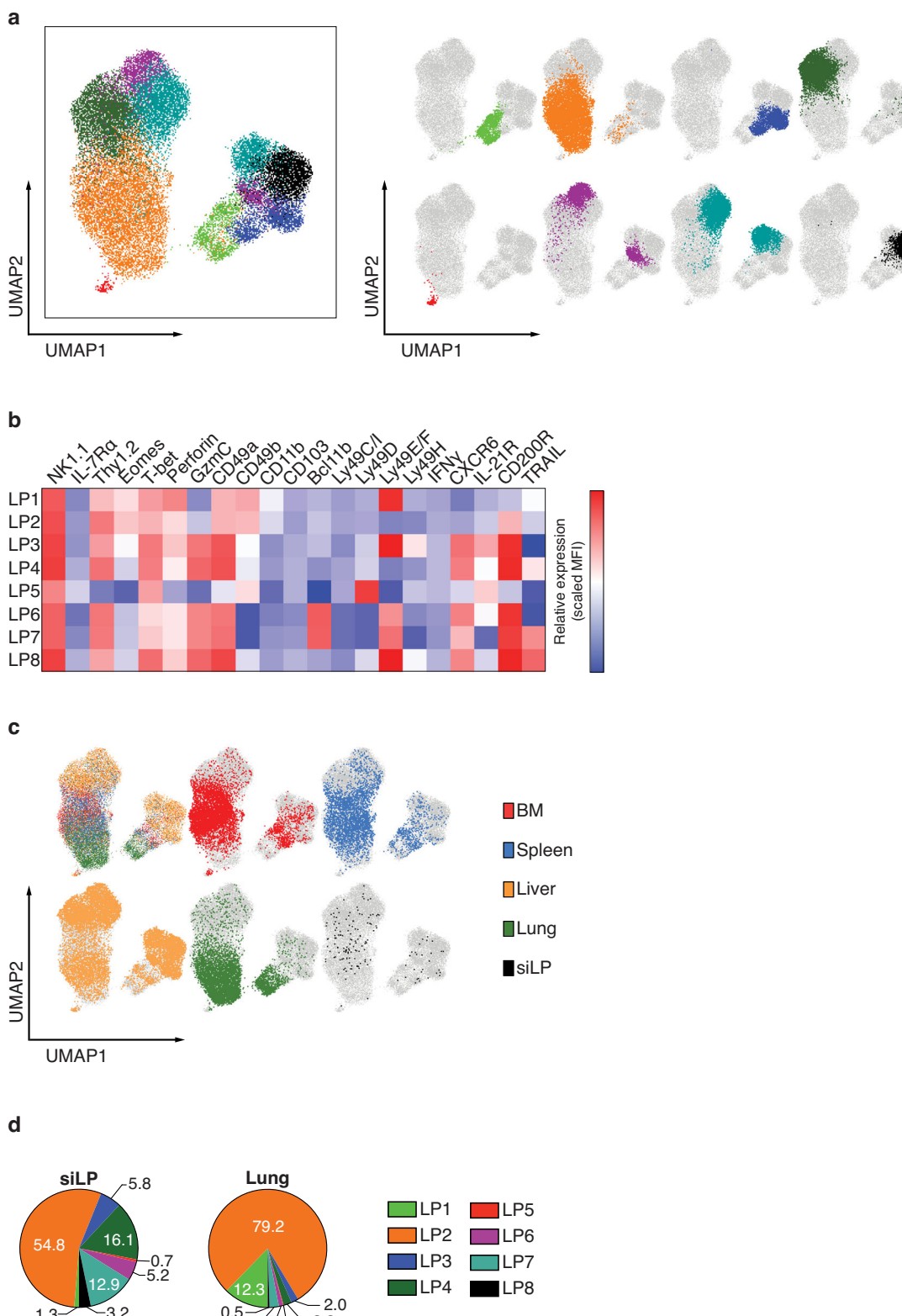

**Fig. 7 | High-dimensional spectral flow cytometry analysis of siLP-ILCP group 1 ILC progeny from several tissues. a** Uniform manifold approximation and projection (UMAP) visualisation of in vivo derived progeny coloured by cluster. **b** Heatmap of expression of the indicated ILC1/NK cell markers within the identified clusters. Colour represents maximum-normalised mean intensity expression of the indicated proteins within each cluster. The relative MFI depicted is a function of both proportion of cells expressing each marker and the protein level of the positive cells within each cluster. **c** UMAP visualisation of in vivo derived progeny coloured by tissue location. **d** Proportion of each identified group 1 ILC cluster within siLP and lung. Data shown is pooled from two independent experiments (5 animals/tissue in total). Source data are provided as a Source Data file.

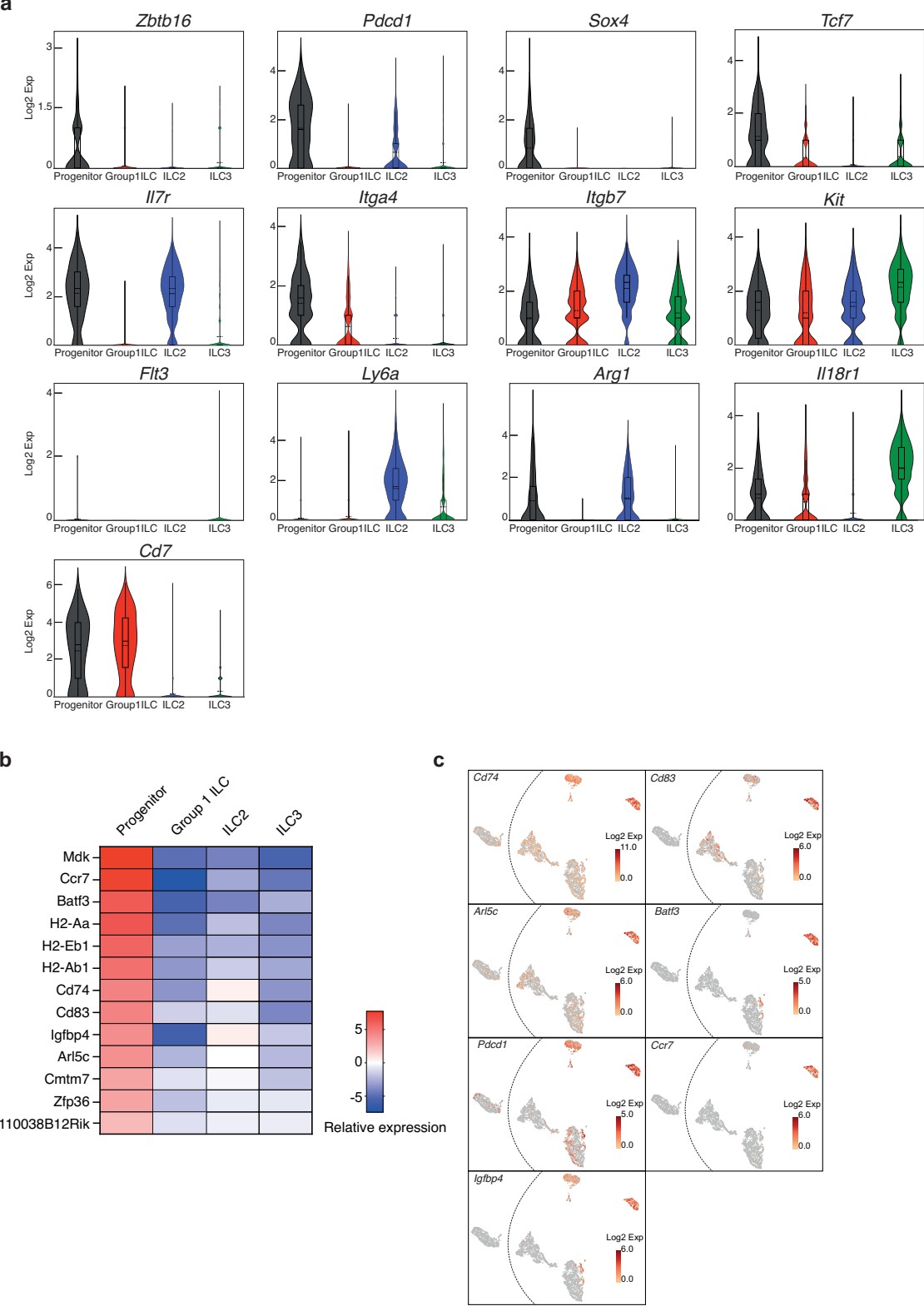

**Fig. 8 | scRNA-seq analysis of siLP-ILCPs shows commonalities with BM-ILCPs and a distinct gene signature from their progeny. a** Violin plots with expression (log2 expression) of indicated genes in Progenitor (375 cells), Group1 ILC (1250 cells), ILC2 (622 cells) and ILC3 (1580 cells) cell populations. Rectangle, solid line, and dashed line represent the interquartile range, the median and the mean, respectively. **b** Heatmap of expression of genes that distinguish the siLP-ILCP progenitors from their progeny in the siLP. **c** UMAP plot with expression level (log2 expression) of indicated genes per individual cell.

transplanted into lymphocyte-deficient mice, whilst their prolific proliferative capacity mirrored that observed in vitro, their developmental profile was more complex and nuanced and the tissue microenvironment radically altered ILC commitment. Similar to the progeny phenotype in vitro those found in the liver, lung and spleen were predominantly restricted to group 1 ILCs. However, in stark contrast, siLP-ILCPs generated all 3 ILC subsets in the recipient intestine, their tissue of origin. The impact of tissue

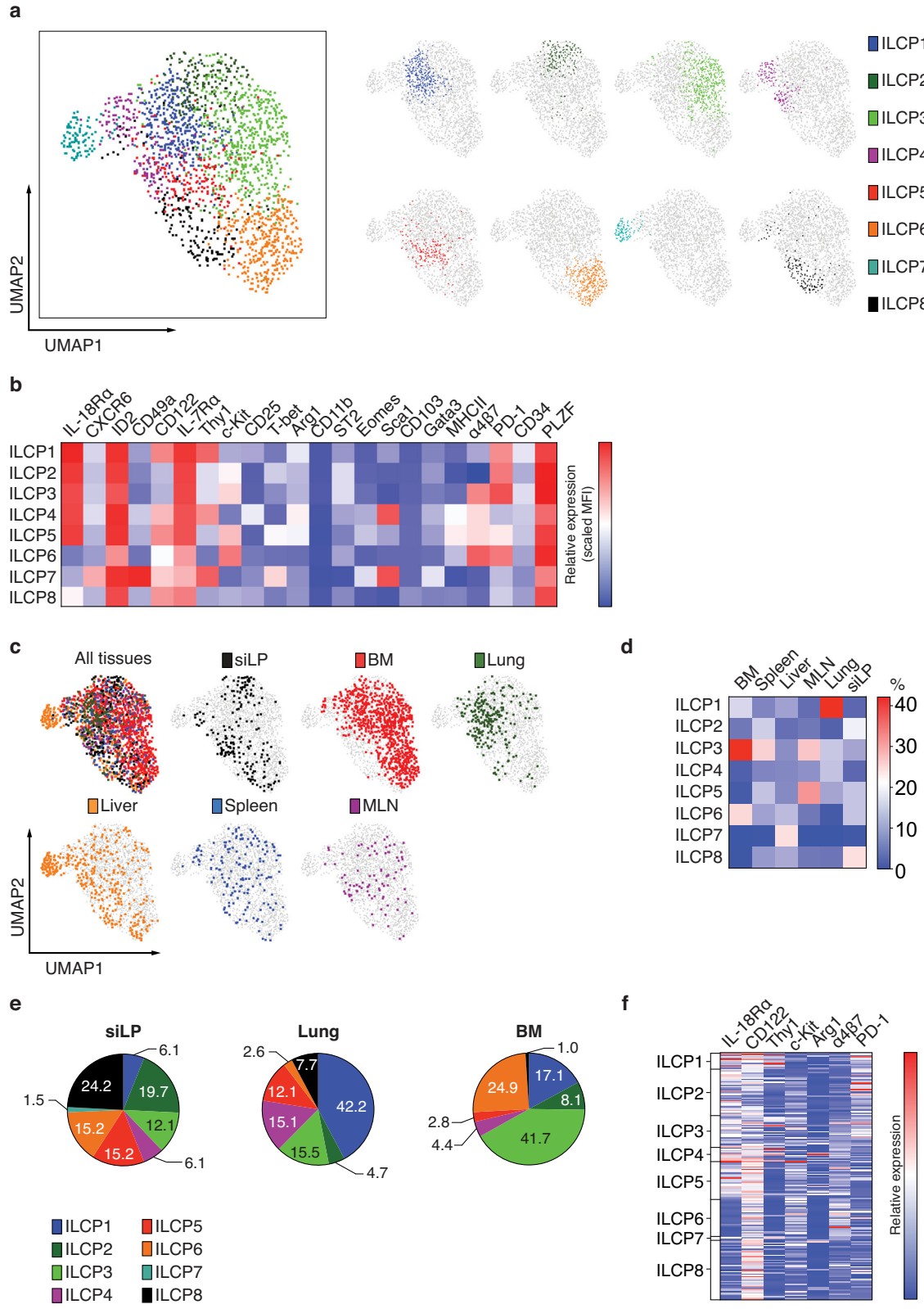

microenvironment was further emphasised by the phenotypic differences we observed between the group 1 ILC progeny in the lung (often with a more NK-like phenotype) and intestine (having a more ILC1-like phenotype). Indeed, detailed phenotypic analysis of the group 1 progeny across multiple recipient tissues was consistent with mounting evidence that group 1 ILCs represent a continuum of phenotypes with an array of transcription factor, surface marker

and cytokine expression which is strongly influenced by tissue location and steady state versus challenged conditions[47–51].

Recent studies have identified bone marrow NK cell progenitors that develop independently of PLZF+ ILCPs[56,57]. Both studies highlight differences between the NK cells derived from their novel NKPs and BM-ILCPs[56,57]. Interestingly, like siLP-ILCPs, other than in the intestine, the recently reported progenitors are biased towards the production

**Fig. 9 | High-dimensional spectral flow cytometry analysis of CD45+lineage−Id2-BFP+IL-7Rα+NK1.1−NKp46−KLRG1−Rorγt-Katushka−CCR6−PLZF+Bcl11b− cells from siLP, BM, spleen, liver, MLN and lung.** Gating strategy for CD45+lineage−Id2-BFP+IL-7Rα+NK1.1−NKp46−KLRG1−Rorγt-Katushka−CCR6−PLZF+Bcl11b− cells shown in Supplementary Fig. 12a. **a** Uniform manifold approximation and projection (UMAP) visualisation coloured by identified subclusters. **b** Heatmap of expression of the indicated markers within the subclusters from (**a**) Colour represents maximum-normalised mean intensity expression of the indicated proteins within each cluster. The relative MFI depicted is a function of both proportion of cells expressing each marker and the protein level of the positive cells within each cluster. **c** UMAP visualisation of siLP-ILCP-like cells coloured by tissue of origin. The first UMAP combines cells from all tissues, each with a different colour, and subsequent UMAPs separate the cells from each tissue and map them onto this UMAP (in grey in the background) to give a representation of their distribution across the UMAP. **d** Heatmap indicating the proportion of each identified siLP-ILCP-like cell cluster within each tissue. **e** Proportion of each identified siLP-ILCP-like cell cluster within siLP, lung and bone marrow. Data shown from one experiment (6 animals/tissue). Source data are provided as a Source Data file. **f** Heatmap of marker expression depicted per individual cell within siLP-ILCPs (198 cells). Colour represents normalised mean intensity expression of the indicated proteins.

of NK cells in vitro and in vivo. However, since the ENKPs described by Ding et al. are Flt3+ and PLZF−[57], and the cells described by Liang et al. are EomeshiNKneg [56], these populations are phenotypically distinct from siLP-ILCPs. Furthermore, whilst the characterised ILCP_NK cells have similarities with the group 1 ILC progeny of the siLP-ILCPs, a number of the genes used to discriminate ENKP_NK cells from ILCP_NK cells are common to the genes which distinguish the siLP-ILCP lung from the siLP progeny[57]. Thus, in the context of our data these studies emphasise the complexities and nuances of group 1 ILC development which is influenced by both progenitor phenotype and tissue location.

This pattern of tissue specific ILC reconstitution from the siLP-ILCPs was distinct from that observed following the transfer of BM-ILCPs which give rise to ILC2s in the lungs of $Rag2^{-/-}Il2rg^{-/-}$ recipients following in vivo transfer[4,33,34] and produce ILC2s in lung organoid cultures[58]. Indeed, in our hands, when we directly compared BM-ILCPs with siLP-ILCPs we found that BM-derived cells were more likely to produce ILC2 progeny in both the lung and intestine. This observation implies that the lungs are not equally permissive to ILC2 development from all ILCPs. Further, in the intestine almost half of the siLP-ILCP progeny were Rorγt+ cells which despite their IL-7Rα+/− status, both at the transcriptional and protein level, had a gene expression profile consistent with bona fide ILC3s. This is consistent with ILC3s forming a much larger proportion of the ILCs in the siLP than in other tissues investigated, under homoeostatic conditions in the mouse[59]. Again, this was in stark contrast to the output of the BM-ILCPs in the intestine. Thus, our data suggest that the microenvironment in the tissue of origin of an ILCP, in this case the siLP, imprints an siLP-specific programme for maintaining progenitor multipotency in the intestine which is lost in other tissues. This would be consistent with the observation from gut and lung organoid studies that the organoid tissue origin exerts a lasting tissue imprint on the ILCs that mature in them[58]. We also noted differential expression of $Cd7$, a marker with ill-defined function in ILCs, by the siLP-ILCPs, which is expressed by BM trans-ILCPs progressing towards a more ILC1/NK/ILC3 trajectory with a loss of ILC2 potential[5]. This may suggest that siLP-ILCPs represent an intestinal pool of CD7-positive transitional ILCPs which retain multipotency in the intestine, with a bias towards ILC3 commitment, but commit to the ILC1/NK lineage in the absence of intestine-specific cues, for example in the lung. In this context, Oherle et al proposed a "bad soil" hypothesis in which an IGF-1-positive pulmonary niche in newborn lungs supports ILC3 development, but that in adult lungs such cues are absent and there are few ILC3s[26].

Our results demonstrate the existence of resident ILC progenitors in the intestines of adult mice with the ability to generate group 1 ILCs, ILC2s and ILC3s. However, upon transfer they predominantly produce only group 1 ILCs in the lungs, spleen and liver, more closely resembling their developmental potential in vitro. Furthermore, siLP-ILCP-like cells from other tissues show subtle inter-tissue variations which raises the possibility that these tissue-educated ILCP may produce tissue-refined ILC progeny. These findings underline the importance of the tissue microenvironment in directing the development of ILCP to specific ILC subsets both at the site of origin of the progenitors and the site of ILC maturation.

## Methods

### Ethics statement
All experiments undertaken in this study were done so with the approval of the MRC-LMB Animal Welfare and Ethical Review Body (AWERB) and of the UK Home Office.

### Mice
All mice were bred in a specific pathogen-free facility. All experiments undertaken in this study were done with the approval of the UK Home Office. $Bcl11b^{tdTom}$ mice were provided by Pentao Liu (Wellcome Sanger Institute, Cambridge, UK)[60]. PLZF-Citrine mice are as previously described[33], Rorγt-Katushka and Id2-BFP mice are as previously described[5]. Four colour reporter mice were a compound strain generated by interbreeding of the individual reporter strains. CD45.1 $Rag2^{-/-}Il2rg^{-/-}$ mice were a gift from James Di Santo and wild type (WT) C57Bl/6JOla mice (Jackson Labs.) were maintained in house. Unless otherwise stated mice were euthanised by rising carbon dioxide concentration.

### Tissue preparation
Cell suspensions of spleen, mesenteric lymph node (MLN) and liver were obtained by passing the tissues through a 70 μm strainer. Lung tissue was pre-digested with 750 U/ml collagenase I (Gibco) and 0.3 mg/ml DNaseI (Sigma-Aldrich) prior to obtaining a single cell suspension. Bone marrow (BM) cells were removed from femurs and tibiae by centrifuging briefly at 6000 × g. For lung, liver, BM and spleen cell suspensions, red blood cells were removed by incubation with RBC lysis solution (140 mM NH4Cl, 17 mM Tris; pH 7.2). Lung lymphocytes were further enriched by centrifugation in 30% Percoll at 800 × g (GE Healthcare) whilst liver lymphocytes were enriched in 40% Percoll at 690 × g.

Blood leucocytes were prepared from 100–500 μl of blood collected in EDTA-treated collection vials. Red blood cells were lysed by incubation with 25 ml RBC lysis solution (140 mM NH4Cl, 17 mM Tris; pH 7.2) for 10 min at room temperature and leucocytes washed and filtered through a 70 μm strainer.

For preparation of siLP and colonic LP lymphocytes, intestinal contents were removed by the application of gentle pressure along the length of the intestine. Intestines were opened longitudinally, cut into 3 cm long pieces and washed briefly by vortexing in PBS + 10 mM HEPES (PBS/HEPES). Epithelial cells were removed by incubation with RPMI supplemented with 2% FCS, 1 mM dithiothreitol and 5 mM EDTA for 2 × 20 min at 37 °C with shaking (200 rpm). Where appropriate small intestine intra epithelial lymphocytes (siIELs) were collected with the epithelial fraction at this point. Intestinal pieces were washed with PBS/HEPES and incubated, with shaking, at 37°C with RPMI + 2% FCS, 0.125 KU/ml DNaseI (Sigma-Aldrich) and 62.5 μg/ml Liberase TL (Roche) until no large pieces of intestine remained. Cells were then passed through a 70-μm strainer, pelleted and separated over a 40%:80% gradient of Percoll at 600 × g for 20 min. LP lymphocytes were isolated from the interface and prepared for flow cytometric analysis. Unless stated otherwise, small intestine and colonic lamina propria (siLP and cLP) includes associated Peyer's patches.

Cell suspensions from adipose tissue were obtained by mechanical dissociation in RPMI-1640, and digested with collagenase I (Life Technologies), DNaseI (Roche) at 37 °C whilst shaking. Initial wash steps were performed with PBS 3% FCS warmed to 37ºC and centrifugation steps (400 × g) were performed at room temperature to allow separation of the cell pellet from the fat.

## Flow cytometry

Single cell suspensions were incubated with fluorochrome-, or biotin-, conjugated antibodies in the presence of anti-CD16/CD32 antibody (Fc block, clone 2.4G2), followed by fluorochrome-conjugated streptavidin where necessary. Antibodies used for flow cytometry and FACS were as detailed in Supplementary Data 2. Biotinylated antibodies were detected with streptavidin conjugated to BUV737 (BD Horizon). All samples were co-stained with a cell viability dye (Fixable dye eFluor780, Invitrogen, or Zombie NIR, Biolegend) and analysis was performed on an LSRFortessa system (BD Biosciences) with FACSDiva software (version 6.2, BD Biosciences) or an ID7000 spectral cytometer (Sony Biotechnology). For cell sorting an iCyt Synergy (70 µm nozzle, Sony Biotechnology) was used. Intracellular transcription factor staining was performed by fixation with 2% PFA for 45 min, followed by incubation with fluorochrome antibodies diluted in perm wash buffer (Foxp3 staining kit, eBioscience). Intracellular cytokine staining was performed by fixation with 2% PFA for 45 min, followed by incubation with fluorochrome antibodies diluted in perm wash buffer (Foxp3 staining kit, eBioscience) following pre-culture with RPMI, supplemented with protein transport inhibitor, for 4 h at 37 °C.

## Flow cytometry data analysis

Flow cytometric analysis was performed using FlowJo, LLC v10 (BD) and associated plugins. For unsupervised dimensionality reduction and clustering, target populations (i.e. siLP-ILCP-derived ILC1/NK progeny (Live CD45.2$^+$Id2$^+$CD11b$^{-/low}$Lin$^-$NK1.1$^+$Rorγ-Kat$^-$ from 5 male and female recipient mice of 11-22 weeks of age), or siLP-ILCP-like cells (as defined in the gating strategy in Supplementary Fig. 12a) from 6 male four colour reporter mice of 16 weeks of age were analysed with X-shift or PhenoGraph for unbiased cluster identification based on the depicted defining markers followed by FlowSOM[61–63]. Uniform manifold approximation and projection (UMAP) visualisation and marker-based heatmaps for identified clusters were generated with UMAP_R and ClusterExplorer plugins in FlowJo. For siLP-ILCP-like population analysis, all samples were concatenated and BM ILCPs were downsampled to 150 events/sample to avoid them from dominating over other tissue-derived ILCP.

## Intravascular CD45 labelling

Intravascular staining for the discrimination of vascular versus tissue resident leucocytes was performed essentially as described by Anderson and colleagues[35]. In brief, each four colour reporter mouse received 3 µg of anti-CD45-APC (clone 30-F11, eBioscience) injected intravenously via the tail vein and 3 min later they were euthanised (cervical dislocation) and tissues harvested for flow cytometric analysis. Male and female mice of 10-24 weeks of age were used.

## OP9 and OP9-DL1 stromal cell co-cultures

OP9 and OP9-DL1[64,65] cells were maintained in complete IMDM (IMDM, supplemented with 20% FCS, 1% penicillin, 1% streptomycin, 0.1% 2-mercaptoethanol and non-essential amino acids (Gibco)). For progenitor cell co-culture, OP9 cells were incubated with 4 µg/ml mitomycin C for 2 h, washed, seeded at a density of $1 \times 10^6$ cells per 96-well plate and allowed time to adhere. Sorted siLP-ILCPs (from male and female four colour reporter mice of 9–30 weeks of age) were seeded onto OP9 monolayers and cultured in complete IMDM, supplemented with 25 ng/ml rmIL-7 (Biolegend) and 25 ng/ml rmSCF (Biolegend), for 3 weeks before flow cytometric analysis of progeny. For comparison of

growth on OP9 versus OP9-DL stromal cells siLP-ILCPs were sorted from male four colour reporter mice of 15–17 weeks of age with a balanced mix of ages in each group.

## Adoptive transfers

Four colour reporter mouse siLP-ILCPs and BM-ILCPs were FACS purified into mouse serum, diluted to 50% with PBS. For any one transfer experiment ILCPs were pooled from several donors of the same sex. Donor mice of both sexes of 7–26 weeks of age were used. Female donor cells were transferred into male and female recipients, male donor cells into male recipients only. Cell suspensions were aspirated with a syringe and implanted via tail vein injection into sublethally-irradiated (450 rad) $Rag2^{-/-}Il2rg^{-/-}$ recipients (male and female, 6–23 weeks of age). Analysis of donor cell progeny was performed 6–7 weeks after cell transfer.

## In vitro stimulation and assay of NK cell effectors

siLP-ILCP progeny were assessed for NK cell effector production by stimulation with IL-2, IL-15 and IL-18 (all at 50 ng/ml) for 24 h for in vivo progeny or 48 h for in vitro progeny followed by flow cytometric analysis for perforin and IFNγ.

## In vitro cytotoxicity assay

B16F10 melanoma target cells (CRL6475 ATCC) were prepared for the assay by staining with CMTPX (AAT Bioquest, 22015), to mark them as red, before plating on 384-well flat bottom tissue culture plate at a density of 60 cells/mm$^2$ and incubating at 37 °C overnight to allow adherence to the 384-well plate. siLP-ILCP progeny from the liver, wildtype splenic NK cells and wildtype splenic CD4$^+$ T cells were incubated overnight in RPMI complete media (RPMI plus 10% FCS, antibiotics, 100 µM 2-mercaptoethanol) with 20 ng/ml rmIL-2 (Biolegend, 575). The following day the siLP-ILCP progeny (FACS purified from male and female recipients of 6–21 weeks of age), NK cells and CD4$^+$ T cells (FACS purified from WT female mice of 11–16 weeks of age) were plated with equivalent effector:target ratios on the prepared B16F10 melanoma cells in RPMI complete media with 20 ng/ml rmIL-2 and 250 nM SYTOX® Green dye (Invitrogen™, S7020). Red and green fluorescence were measured every 2 h using an IncuCyte (Sartorius) and cell death of B16F10 cells was quantified by the area of red-green overlap, using a size cut-off to exclude lymphocytes.

## Single-cell RNA sequencing

A 10× single-cell library preparation was performed using the 10x Genomics technology platform. The 10x Genomics Chromium Single Cell 3′ v3 protocol was followed to obtain 3′ libraries for subsequent sequencing. The reads were aligned to the mouse transcriptome (GRCm38), and expression was calculated using the 10× Cell Ranger (version 6.0.1) wrapper for the STAR aligner (version 2.7.2a). Separate libraries were generated from the siLP-ILCPs purified by FACS from the four colour reporter mice (male and female 8–16 weeks of age) as described in Supplementary Fig. 1 (PLZF$^+$Bcl11b$^-$ sub-population, 1048 cells), the siLP progeny (9098 cells) (female siLP-ILCP recipient mice of 17 weeks of age) and the lung progeny (6045 cells) (male and female siLP-ILCP recipient mice of 6 to 8 weeks of age) FACS sorted as CD45.2$^+$Id2-BFP$^+$. These libraries were then combined using Cell Ranger with the addition of previously reported[33] (GSE213814) data from the lung progeny of the BM aceNKPs (2256 cells). Cells were manually clustered based on the UMAP due to the clear and obvious separation. This clustering concurred with both of default Cell Ranger k-means and graph-based clustering methods. Analysis and statistical calculations were performed using the 10x Genomics Loupe Browser (https://support.10xgenomics.com/single-cell-gene-expression/software/visualization/latest/what-is-loupe-cell-browser). Heatmaps of expression were drawn using the R heatmap function with cells clustered using the default hclust function (hierarchical clustering).

## Statistical analysis

Statistical analysis was performed using GraphPad Prism v10.1.1 software. Data are plotted as mean with SEM error bars. Statistical tests used are detailed in respective figure legends.

## Reporting summary

Further information on research design is available in the Nature Portfolio Reporting Summary linked to this article.

## Data availability

All sequencing data generated in this study have been deposited with the Gene Expression Omnibus (GEO) under accession number GSE234835. Previously published sequencing data for the aceNKP progeny are from GSE213814. All other data are available in the article and its Supplementary files or from the corresponding author upon request. Source data are provided with this paper.

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

## Acknowledgements

We are grateful to the Ares staff, genotyping facility and flow cytometry core for their technical assistance, in particular Claire Knox, Jasmine Farnsworth, Jennifer Roe and Fan Zhang. We are grateful to Pentau Liu for providing the Bcl11b-tdTomato mice and James Di Santo for the $Rag2^{-/-}Il2rg^{-/-}$ mice. Funding: this study was supported by grants from the UK Medical Research Council (U105178805) (MG, NRR, JEM, HEJ, ANJM) and Wellcome Trust (100963/Z/13/Z and 220223/Z/20/Z) (PAC, ACFF, MG, JAW). European Union's Horizon 2020 research and innovation program under the Marie Skłodowska-Curie grant agreement number 896454 (NRR). Rosetrees Trust (JEM). For the purpose of open access, the MRC Laboratory of Molecular Biology has applied a CC BY public copyright licence to any Author Accepted Manuscript version arising.

## Author contributions

P.A.C designed and performed experiments and wrote the paper. M.G., N.R.R, A.C.F.F., J.E.M., J.A.W., A.C., H.E.J. performed experiments, provided advice on experimental design and interpretation, and commented on the manuscript. J.D.S. provided reagents and commented on the manuscript. A.N.J.M. supervised the project, designed the experiments and wrote the paper.

## Competing interests

The authors declare no competing interests.
