## [Peer Review File · Nature Communications]

Recipient tissue microenvironment determines developmental path of intestinal innate lymphoid progenitorsREVIEWER COMMENTS

Reviewer #1 (Remarks to the Author):

Clark et al. use a unique combination of “polychromatic” transcription factor reporter mice to identify progenitors of innate lymphoid cells (ILCP) in the small intestine of adult mice. They use single-cell cultures and transplantation into lymphopenic mice to establish the differentiation potential of these ILCP. Interestingly, they find that the subtypes of ILCs emerging as progeny of these ILCP is determined by the tissue, in which they engraft. In lungs and livers, for example, the ILCP generate NK cells and ILC1, but little to no ILC2 and ILC3. In the intestine, in contrast, ILC2 and ILC3 are efficiently generated. By comparing their results to previous studies, which showed that transplanted ILCP from BM can actually generate ILC2 and ILC3 in lung and livers and lungs, the authors conclude that the SI-ILCP is more restricted in its developmental potential, and depends more on the local tissue-environment. This is an elegant and well-conducted study reporting novel and interesting findings.

Major criticism:

- The presented experiments suggest that the developmental potential of SI-ILCP in the liver and lung is limited, compared to BM-ILCP. This comparison is, however, not experimentally performed, and BM vs SI-ILCP are not tested side-by-side. Authors instead refer to previous publications, but it would be important to perform side-by-side analyses to show that in the differentiation potential of BM-ILCP and SI-ILCP indeed differs in both, the specific culture conditions as well as in the recipient animals used here. Optimally, such transfers could be performed with congenically marked BM- vs SI-ILCP in the same recipient, if the polychromatic system allows for it.
- Previous tissue-ILCP have been reported in lungs, and ILC1-progenitors in the liver. Authors should apply the gating strategies used in those papers to map those cells with their elegant polychromatic reporter system. How similar are these progenitor populations in adult tissues? Such comparisons (similar to what they already do for the embryonic ILCP by Bando et al., and in part by testing I118r expression for lung ILCP by Zeis et al. and Ghaedi et al. -please also cite this paper) would be very informative. Also, the authors should use their elegant system to test whether they find similar candidate ILCP in other tissues: Are such cells present only in the intestine or also the colon? Only in lamina propria or also IEL? How about the adipose? What about the spleen or lymph nodes, e.g. the mesenteric ones, in which ILC have been proposed to be tissue-resident as well? I don't think the authors need to do any validation of progenitor potential for these cells, but applying their unique and established system for few more organs would provide valuable insights for the community, and increase the impact of the manuscript.

Minor points:

- Figure 5b: Which cells are analyzed as progeny? All three clusters shown for siLP in 5a?
- Authors suggest that siLP-ILCP-derived ILC1/NK cell progeny found in the siLP have a more tissue-resident phenotype than those located in the lungs, but the genes mentioned map to ILC1 (Cxcr6, Emb, Asb2, Il21r) versus cNK (Cma1, Itgam, Zeb2) – likely this reflects different % of ILC1 versus NK cells in the progeny in the lung versus the intestine? Reflecting much increased cNK over ILC1 in the lungs of SPF mice, and vice versa in the SI-PL? Or do the authors really find evidence for differential expression of “residency marker genes” in the same cell types (ILC1 vs ILC1, cNK vs cNK)? I feel this should be validated (could be done by FACS) and clarified as it is even stated in the abstract as a major finding of the manuscript.
- “Our data also suggest that the microenvironment in the tissue of origin of an ILCP, in this case the siLP, imprints an siLP specific programme for maintaining progenitor multipotency in the intestine which is lost in other tissues.” – in the absence of a direct experimental comparison (e.g. cotransfer) of a multipotent (BM) ILCP, I find it very difficult to make this statement. This should be experimentally tested (see above).

-

- "siLP-ILCPs expressed CD7, which, was not expressed by BM-ILCP 5," – is Cd7 a distinguishing feature of BM ILCP versus tissue-ILCP across studies? Or are there controversial data for BM-ILCP?

- "the tissue residence of siLP-ILCPs sets them apart from lung ILCPs which were predominantly labelled by ivCD45 implying their recent recruitment from the circulation 20. " – I understood from that study that, based on parabiosis, there were both resident and circulating progenitors? And that even ivCD45+ ILCP did not fully equilibrate in parabiotic mice?

- "The expression of the Rorgt reporter in the siLP-localised progeny of the siLP-ILCPs and their scRNAseq analysis confirmed them as bona fide ILC3s. However, we did observe a sub-population of these cells with low to no IL-7Ra expression." – by FACS only, or also at the mRNA level? Was IL7ra stained intracellularly, which may detect downregulated receptor? Please clarify.

Reviewer #2 (Remarks to the Author):

The current manuscript by Clark et al. uses a tetra-combinatorial reporter mouse line for Id2, Rorc, Bcl11b, and Zbtb16 to describe a putative ILC progenitor population in the small intestinal lamina propria of adult mice. Using in vitro differentiation cultures and in vivo adoptive transfers, these siLP-ILCP (Id2+Rorc-Kat-Bcl11b-Tom-Zbtb16-Cit+) preferentially differentiate towards group 1 ILC lineages while the intestinal tissue niche additionally generates ILC2 and ILC3.

In mice, ILCPs have been described in the adult bone marrow and lung but also in the fetal liver, fetal lymph node anlagen, and the fetal intestine. Hence, the finding of ILCPs in the adult intestine represents a novel finding important for the field; however, as it stands, the potential and full characterization of the siLP-ILCP is incompletely studied and would need further investigation. I have the following remarks:

Major comments

1. In-depth characterization of siLP-ILCP in tetra-combinatorial reporter mice is missing

The data of the current manuscript identifies an ILCP population specifically in the small intestinal lamina propria of adult mice. By excluding mature ILC lineages using the markers NKp46, NK1.1, and KLRG1 as well as RORgt-Katushka, the authors describe several progenitor populations based on Bcl11b-Tom and PLZF-Cit expression (Figure 1b, Figure S1). To confirm their ILCP identity and more convincingly exclude the possible contamination of lineage committed cells, the authors should additionally show that these populations do not express any other transcription factor such as T-bet (or GATA3hi). Besides this, Figure S3 aims at identifying these progenitors in other tissues such as spleen and mLN, as well as in blood. Also here the flow cytometric approach using a minimal set of surface markers to exclude mature ILC lineages should be extended. In addition, the manuscript would greatly benefit from a comparison of the peripheral ILCPs to the bone marrow ILCPs in the tetra-combinatorial mice in both, in vitro and in vivo assays.

2. Incomplete study of all si-ILCP

Based on the in vitro proliferation data, the authors exclusively focuses on PLZF+ Bcl11b- ILCPs in the experimental systems (Figure 2-6) of the current manuscript, while fail to analyze other subsets, especially the PLZF+ Bcl11b+ subset. It is unclear whether low proliferation of some subsets is due to intrinsic properties or to suboptimal culture conditions. The in vitro culture system could be optimized and earlier time points than 3 weeks could be analysed, as most ILCP cultured rather generate ILC1. Moreover, the differentiation potential of the PLZF+ Bcl11b+ subset should be tested in vivo.

3. Quality of the data and analysis

The flow cytometric analysis should be revisited (see also specific points to each Figure further down) and thus question the ILCP sorting and subsequent findings.

Specific points

- Figure 1: it would be of importance to compare the in vitro differentiation potential of intestinal progenitor populations to the potential of bone marrow-derived precursors on OP9 but also OP9-DL1 feeder cells. Figure 1 c-e only depicts the readouts for the P+B- precursor population. Which cells could be generated from the P+B+, P-B+ and P-B- progenitors?
- Figure 3: It would be important to also analyze bone marrow in the in vivo adoptive transfer setting to see if ILCPs could also relocate to the bone marrow. Furthermore, inclusion of a control population would help to identify positive and negative populations in the flow cytometric representatives (NKp46 vs NK1.1 and Eomes vs NK1.1 plots). Besides mature ILC lineages, can ILCPs still be detected?
- Figure 5: a, the dimensionality reduction (UMAP) should be indicated in the plot. In addition, it would be relevant to set mature ILC-related markers (Robinette et al., PMID: 25621825; Gury-BenAri et al., PMID: 27545347) into context.
- Figure 6: How are lineage determining transcription factors such as GATA3 and T-bet expressed throughout the scRNAseq dataset? A) is lacking the unit in the color legend
- Figure S1: ILCs, especially ILC3, are described to express lower levels of CD45 (e.g., Zhou et al. PMID: 30944470 Ext. Data 3). Would inclusion of cells expressing CD45 intermediately increase the ILC(P) populations? Following the gating strategy, the lineage gating is not very convincing as it is hard to evaluate positive and negative cells, maybe plotting lineage vs. Id2-BFP or IL-7Ra would help to identify a negative lineage. Similarly, the KLRG1-CCR6-NKp46- gate is unclear, maybe plotting APC/EF660 vs RORgt-Kat would help to identify a negative population. In particular, gating of Bcl11b-Tomato vs PLZF-Citrine should be revised and question the purity of sorted populations for the in vitro assays in Figure 1c-e.
- Figure S3: Can ILCPs based on Plzf and Bcl11b also be identified in other organs analyzed (gating adjusted to tissue-specific ILC markers)?
- Figure S6: It is very hard to draw any conclusion from the data as they are presented. A heatmap showing the gene expression pattern per cluster would help to see differences and overlapping genes between siILCP and progeny. Expression of selected genes as violin plot would help the reader to evaluate overall expression within the individual clusters.
- Figure legends: n should always be stated
- Discussion: "Although we identified expression of the IL-25 receptor by some siLP-ILCPs, in vitro inclusion of IL-25 in siLP-ILCPs cultures, which is produced by the gut epithelium (but less so in the lung) and induces ILC2 expansion, did not alter their differentiation." Please provide a reference for IL-25 levels in lungs vs intestine.
- Several important citations about human and murine tissue ILCP are missing. Please revisit the bibliography and include at least:
 - Chea et al., PMID: 26832410: mouse fetal spleen and mLN ILCP
 - Ghaedi et al., PMID: 3181-6636: murine adult and neonatal lung
 - Liu et al., PMID: 34239074: human fetal tissues from 8, 10, and 12 weeks PCW, specifically the liver, thymus, spleen, intestine, skin, and lung
 - Montaldo et al., PMID: 25500367, human adult tonsil CD34+ RORgt+ progenitors
 - Scoville et al., PMID: 27178467, human secondary lymphoid tissue CD34+ RORgt+ ILCP
 - Simic et al., PMID: 32783932: mouse fetal lymph node ILCP
 - Stehle et al., PMID: 34556887: mouse fetal intestine ILCP
 - Suo et al., PMID: 35549310, human prenatal intestine scRNA ILCP

Reviewer #3 (Remarks to the Author):

In this manuscript, Clark et al. employed a multicolor fluorescent reporter mouse model, for Id2, PLZF, Bcl11b and Rorgt, to identify ILC progenitors residing in the small intestine. Next, the ability of these precursor cells to generate distinct NK/ILC lineages was assessed using in vitro and in vivo systems. The authors found a siLP-ILCP population able to give rise to a distinct NK/ILC lineages in the small intestine, while the potential to generate NK/ILCs in other tissues, namely in spleen, liver and lung was limited to Eomes+ cells.

The manuscript is interesting and helps fuel the current view of ILC-poiesis in the tissues. Some

issues emerged that need to be clarified (discussed below).

1. A small fraction of siLP Gata3bright ILC2 does not express KLRG1. Is it possible that some of the lin- Id2+IL7R+NKp46-NK1.1-KLRG1-Rorgt-CCR6- cells shown in Figure 1b contain these KLRG1-Gata3bright ILC2? This might explain the enrichment of Bcl11b-Tomato+ cells in the contour plot, or the results presented in Figure 4.

2. Are siLP-ILCP found in the large intestine?

3. Related to Figure 3A. siLP-ILCP population seems to give rise mainly Eomes+ NK cells in spleen, lung and liver. Do siLP-ILCP lose the capacity to generate Eomes- ILC1 in vivo?

4. Moreover, the small intestine is generally enriched in Eomes- ILC1. Can siLP-ILCP generate Eomes- ILC1 in this tissue? Few more markers helping characterize the phenotype of the ILC1/NK cells generated in vivo from the siLP-ILCP are needed (CD49a, CD49b, Ly49s and others), at least in one representative tissue.

5. Figure S6. To evaluate signs of early differentiation in siLP-ILCP, the authors should use gene signatures from an independent dataset (Immgen?). The current analysis is quite circular and the authors might lose important information. In alternative, siLP-ILCP should be excluded when defining ILC-specific signatures.

6. The scRNA-seq data could be further analyzed to be more informative. Is there any combination of surface markers/molecules helping to gate siLP-ILCP without using reporter mice? Is it possible to define the signature of siLP-ILCP and look for these cells in human gut datasets? These infos would be precious for further studies.

Minor

- Page 4; lanes 99-101. Citation or FACS plot should be added at the end of this sentence.

- Figure 2b. I feel that keeping lin+ cells in the main Figure is confusing. Moreover, I feel the authors should choose just one way to represent the flow data (contour plot?) in the same figure, for consistency.

- Information regarding scRNA-seq analysis is very limited.

Responses to reviewers

We thank the reviewers for their comprehensive and helpful evaluation of our paper. To address the criticisms raised by the reviewers we have now performed several new mouse experiments (apologies for the delay, but we had to expand several colonies to produce the mice required). To improve the quality of the results we have now used spectral flow cytometry to demarcate progenitors and progeny in multiple tissues. Taking on-board the suggestion of the reviewers we have also re-analysed the scRNA-seq data to improve the resolution and presentation of these results. The manuscript has essentially been rewritten, with the inclusion of large amounts of new and re-analysed data and changes to the order of the sections. These changes, made in response to insightful suggestions by the reviewers, have greatly improved the quality of our paper, and we hope it is now acceptable for publication in *Nature Communications*.

Reviewer #1 (Remarks to the Author):

Clark et al. use a unique combination of “polychromatic” transcription factor reporter mice to identify progenitors of innate lymphoid cells (ILCP) in the small intestine of adult mice. They use single-cell cultures and transplantation into lymphopenic mice to establish the differentiation potential of these ILCP. Interestingly, they find that the subtypes of ILCs emerging as progeny of these ILCP is determined by the tissue, in which they engraft. In lungs and livers, for example, the ILCP generate NK cells and ILC1, but little to no ILC2 and ILC3. In the intestine, in contrast, ILC2 and ILC3 are efficiently generated. By comparing their results to previous studies, which showed that transplanted ILCP from BM can actually generate ILC2 and ILC3 in lung and livers and lungs, the authors conclude that the SI-ILCP is more restricted in its developmental potential, and depends more on the local tissue-environment. This is an elegant and well-conducted study reporting novel and interesting findings.

Major criticism:

- The presented experiments suggest that the developmental potential of SI-ILCP in the liver and lung is limited, compared to BM-ILCP. This comparison is, however, not experimentally performed, and BM vs SI-ILCP are not tested side-by-side. Authors instead refer to previous publications, but it would be important to perform side-by-side analyses to show that in the differentiation potential of BM-ILCP and SI-ILCP indeed differs in both, the specific culture conditions as well as in the recipient animals used here. Optimally, such transfers could be performed with congenically marked BM- vs SI-ILCP in the same recipient, if the polychromatic system allows for it.

We now include a side-by-side comparison of the progeny emergent from recipients of either bone marrow-ILCPs or siLP-ILCPs (produced from the same donor animals) to confirm the contrasting outcomes in our hands. Carrying this out in the same recipient animals was not practical because of the enormous amount of time that would have been required to intercross the reporter strains onto a CD45.1 background (or another constitutive reporter etc) to generate the necessary animals. These new data are now presented in Figure 4d and referred to in Lines 214-223.

- Previous tissue-ILCP have been reported in lungs, and ILC1-progenitors in the liver.

Authors should apply the gating strategies used in those papers to map those cells with their elegant polychromatic reporter system. How similar are these progenitor populations in adult tissues? Such comparisons (similar to what they already do for the embryonic ILCP by Bando et al., and in part by testing Il18r expression for lung ILCP by Zeis et al. and Ghaedi et al. -please also cite this paper) would be very informative.

We appreciate this suggestion. However, replication of the gating strategies used by others would be challenging (e.g. the reporters we use preclude use of exact antibody matches; the Locksley paper uses a Arg-reporter and a Rorc fate-mapper; in some instances antigen challenges have been used; etc). However, we have addressed the reviewer's question by applying our gating strategy across a number of adult tissues and phenotyped the populations identified by high-dimensional spectral analysis. In our results we compare the phenotypes of our population across those tissues with the reports the reviewer has highlighted. Our new data are presented in Figure 9 and referred to in Lines 344-367.

Also, the authors should use their elegant system to test whether they find similar candidate ILCP in other tissues: Are such cells present only in the intestine or also the colon? Only in lamina propria or also IEL? How about the adipose? What about the spleen or lymph nodes, e.g. the mesenteric ones, in which ILC have been proposed to be tissue-resident as well? I don't think the authors need to do any validation of progenitor potential for these cells, but applying their unique and established system for few more organs would provide valuable insights for the community, and increase the impact of the manuscript.

We thank the reviewer for this suggestion to improve the manuscript. We do find this population in the spleen and MLN, but not the Fat or IEL. We now include high dimensional spectral analysis of our population from a number of tissues and report a variety of subtle sub-phenotypes. These new data are shown in Figure 9 and S12, and referred to in Lines 344-367.

Minor points:

- Figure 5b: Which cells are analyzed as progeny? All three clusters shown for siLP in 5a?

In the original manuscript the cells analysed as progeny were all three clusters combined. However, in the light of the reviewer's comment we have revisited the presentation of the data and now separate out the 3 progeny subtypes for clarity. These data are shown in Figure 8a and referred to in Lines 323-339.

- Authors suggest that siLP-ILCP-derived ILC1/NK cell progeny found in the siLP have a more tissue-resident phenotype than those located in the lungs, but the genes mentioned map to ILC1 (Cxcr6, Emb, Asb2, Il21r) versus cNK (Cma1, Itgam, Zeb2) – likely this reflects different % of ILC1 versus NK cells in the progeny in the lung versus the intestine?

Our scRNAseq data analysis highlighted that genes previously associated with for type-1 ILCs being tissue-specific or non-tissue-specific were amongst those most differentially expressed between our type-1 siLP and lung progeny. This could be interpreted as reflecting

a different proportion of ILC1 vs NK cells but other markers, thought to distinguish these cell types, did not correlate with this. Indeed, both our scRNAseq and high dimensional spectral analysis of the phenotype of type-1 ILC progeny indicate that they do not align with a simple ILC1/NK cell dichotomy. Rather, using a range of published markers suggested to be useful in distinguishing these cell types, we find that across different tissues there is a variety of type-1 ILC phenotypes more akin to an ILC1/NK continuum. These new data are shown in Figure S8 and referred to Lines 259-271; Figure 6 and referred to Lines 272-277; Figure 7; Figure S9 and referred to Lines 279-304.

Reflecting much increased cNK over ILC1 in the lungs of SPF mice, and vice versa in the SI-PL? Or do the authors really find evidence for differential expression of “residency marker genes” in the same cell types (ILC1 vs ILC1, cNK vs cNK)? I feel this should be validated (could be done by FACS) and clarified as it is even stated in the abstract as a major finding of the manuscript.

As in the previous response, these new data are shown in Figure S8 and referred to Lines 259-271; Figure 6 and referred to Lines 272-277; Figure 7; Figure S9 and referred to Lines 279-304.

- “Our data also suggest that the microenvironment in the tissue of origin of an ILCP, in this case the siLP, imprints an siLP specific programme for maintaining progenitor multipotency in the intestine which is lost in other tissues.” – in the absence of a direct experimental comparison (e.g. cotransfer) of a multipotent (BM) ILCP, I find it very difficult to make this statement. This should be experimentally tested (see above).

-

As above, we now include a side-by-side comparison of the progeny emergent from recipients of either bone marrow-ILCPs or siLP-ILCPs (produced from the same donor animals) to confirm the contrasting outcomes in our hands. Carrying this out in the same recipient animals was not practical because of the enormous amount of time that would have been required to intercross the reporter strains onto a CD45.1 background (or another constitutive reporter etc) to generate the necessary animals. These new data are now presented in Figure 4d and referred to in Lines 214-223.

- “siLP-ILCPs expressed CD7, which, was not expressed by BM-ILCP 5,” – is Cd7 a distinguishing feature of BM ILCP versus tissue-ILCP across studies? Or are there controversial data for BM-ILCP?

We apologise for a lack of clarity - our previous data suggested that BM-ILCPs with a more type1/3 trajectory express CD7. The text has been amended to correct this (Lines 338-339 and Lines 455-461).

- “the tissue residence of siLP-ILCPs sets them apart from lung ILCPs which were predominantly labelled by ivCD45 implying their recent recruitment from the circulation 20. “ – I understood from that study that, based on parabiosis, there were both resident and circulating progenitors? And that even ivCD45+ ILCP did not fully equilibrate in parabiotic mice?

We thank the reviewer for alerting us to this error. The text is now amended to reflect that a significant proportion but not all of the lung ILCs are ivCD45 labelled (Lines 416-418).

- “The expression of the Rorgt reporter in the siLP-localised progeny of the siLP-ILCs and their scRNAseq analysis confirmed them as bona fide ILC3s. However, we did observe a sub-population of these cells with low to no IL-7Ra expression.” – by FACS only, or also at the mRNA level? Was Il7ra stained intracellularly, which may detect downregulated receptor? Please clarify.

We have not analysed internalised IL7R α , however our scRNAseq data suggest that the gene expression is also down-regulated and we include reference to this in the discussion. We offer a possible explanation for our observation of uncharacteristic low-to-absent IL7R α expression. The cells which are clustered as ILC3s by the scRNAseq data express many other genes that confirm them as bona fide ILC3s (Figure 5d, Lines 203-207 and Lines 446-450).

Reviewer #2 (Remarks to the Author):

The current manuscript by Clark et al. uses a tetra-combinatorial reporter mouse line for Id2, Rorc, Bcl11b, and Zbtb16 to describe a putative ILC progenitor population in the small intestinal lamina propria of adult mice. Using in vitro differentiation cultures and in vivo adoptive transfers, these siLP-ILC (Id2+Rorc-Kat-Bcl11b-Tom-Zbtb16-Cit+) preferentially differentiate towards group 1 ILC lineages while the intestinal tissue niche additionally generates ILC2 and ILC3.

In mice, ILCs have been described in the adult bone marrow and lung but also in the fetal liver, fetal lymph node anlagen, and the fetal intestine. Hence, the finding of ILCs in the adult intestine represents a novel finding important for the field; however, as it stands, the potential and full characterization of the siLP-ILC is incompletely studied and would need further investigation. I have the following remarks:

Major comments

1. In-depth characterization of siLP-ILC in tetra-combinatorial reporter mice is missing. The data of the current manuscript identifies an ILC population specifically in the small intestinal lamina propria of adult mice. By excluding mature ILC lineages using the markers NKp46, NK1.1, and KLRG1 as well as RORgt-Katushka, the authors describe several progenitor populations based on Bcl11b-Tom and PLZF-Cit expression (Figure 1b, Figure S1).

We now present amended and additional analysis of our scRNAseq data for the siLP-ILCs which we hope gives the reader better insight into the phenotype of these cells. Further, we include new results detailing high dimensional spectral analysis of the siLP-ILCs (new Figure 5c and Lines 310-313; new Figure S7 and Lines 313-314; Figure 8 and Lines 321-342; new Figure 9; new Figure S12 and Lines 344-367).

To confirm their ILC identity and more convincingly exclude the possible contamination of lineage committed cells, the authors should additionally show that these populations do not express any other transcription factor such as T-bet (or GATA3hi).

We thank the reviewer for this suggestion. We have now included an additional gene expression figure (S10) that shows that the siLP-ILCPs express little or no Tbx21 (T-bet), nor are they Gata3 high which is clear when compared to the gene expression level of the ILC2 cluster. Further Figure 5d shows little to no Eomes expression, and our high dimensional spectral analysis of the siLP-ILCPs confirms these gene expression data at the protein level (Figure S10 and Lines 315-319).

Besides this, Figure S3 aims at identifying these progenitors in other tissues such as spleen and mLN, as well as in blood.

This figure was intended to demonstrate the gating strategy to identify ILCs in the four tissues and our ILCP population in the siLP for the purposes of measuring the extent of their intra-vascular staining by α CD45 and not to identify ILCPs in other tissues. We recognise this is not clear from the title of the figure legend and this is now amended (Lines 839-841). However, we have now done a survey of a number of tissues for cells that fall into our ILCP gate and phenotyped them using high dimensional spectral analysis (new Figure 9; Figure S12 and Lines 344-367).

Also here the flow cytometric approach using a minimal set of surface markers to exclude mature ILC lineages should be extended.

We believe that the use of our unique combination of reporters and surface markers are sufficient to exclude all mature ILC lineages. All type-1 ILCs will express NK1.1 and Nkp46. As well as excluding most ILC2s KLRG1 will exclude NK cells. Bcl11b is a requirement for ILC2 commitment and therefore excluding those cells positive for this will remove any cells on an ILC2 trajectory. Ror γ t is the archetypal ILC3 transcription factor and thus exclusion of cells expressing the Katushka reporter would remove ILC3s. Further our subsequent analysis of the gene and protein expression patterns of other classical markers of mature ILCs indicates little to no expression of other key markers such as T-bet, Eomes and ST2 and low expression of Gata3. See new Figure S10 and Lines 315-319; and new Figure 9).

In addition, the manuscript would greatly benefit from a comparison of the peripheral ILCPs to the bone marrow ILCPs in the tetra-combinatorial mice in both, in vitro and in vivo assays.

We now include a side-by-side comparison of the progeny emergent from recipients of either bone marrow-ILCPs or siLP-ILCPs (produced from the same donor animals) to confirm the contrasting outcomes in our hands. Carrying this out in the same recipient animals was not practical because of the enormous amount of time that would have been required to intercross the reporter strains onto a CD45.1 background (or another constitutive reporter etc) to generate the necessary animals. These new data are now presented in Figure 4d and referred to in Lines 214-223.

2. Incomplete study of all si-ILCP

Based on the in vitro proliferation data, the authors exclusively focuses on PLZF⁺ Bcl11b⁻ ILCPs in the experimental systems (Figure 2-6) of the current manuscript, while fail to analyze other subsets, especially the PLZF⁺ Bcl11b⁺ subset. It is unclear whether low proliferation of some sbsets is due to intrinsic properties or to suboptimal culture conditions. The in vitro culture system could optimized and earlier time points than 3 weeks

could be analysed, as most ILCP cultured rather generate ILC1. Moreover, the differentiation potential of the PLZF+ Bcl11b+ subset should be tested in vivo.

We hypothesised that the population most likely to include ILC progenitors was the PLZF-positive population and those that were Bcl11b positive were likely already committed to the ILC2 lineage. Our initial in vitro analysis confirmed that the only population to expand to any extent under neutral conditions was indeed the P+B- population. We specifically chose these conditions to avoid skewing the outgrowth in favour of any particular ILC sub-type. Also, the culture duration was largely dictated by the very small numbers of cells isolated from each mouse and the requirement to wait for sufficient progeny to emerge to permit meaningful phenotypic analysis. In the light of these proliferation data, we concentrated our efforts on the P+B- population as they were not showing markers indicative of ILC2 commitment.

3. Quality of the data and analysis

The flow cytometric analysis should be revisited (see also specific points to each Figure further down) and thus question the ILCP sorting and subsequent findings.

We believe our sorting strategy is valid and we have clarified our sorting strategy figures to justify this. We have also additionally validated these data as described above using spectral flow cytometry and gene expression analysis (Figure S1; new Figure 5d; new Figure 9; new Figure S10 and Lines 315-319).

Specific points

- Figure 1: it would be of importance to compare the in vitro differentiation potential of intestinal progenitor populations to the potential of bone marrow-derived precursors on OP9 but also OP9-DL1 feeder cells.

We now include data demonstrating that growth of our progenitors on OP9-DL1 stromal cells makes no difference to the expansion or phenotype of the progeny that grow in vitro. We have addressed the comparison with BM ILCPs in vivo as we believe the outcome in this setting is more meaningful (Figure S3 and Lines 149-151: and Figure 4d).

Figure 1 c-e only depicts the readouts for the P+B- precursor population. Which cells could be generated from the P+B+, P-B+ and P-B- progenitors?

We hypothesised that the population most likely to include ILC progenitors was the PLZF positive population and those that were Bcl11b positive were likely already committed to the ILC2 lineage. Our initial in vitro analysis confirmed that the only population to expand to any extent under neutral conditions was indeed the P+B- population. We therefore concentrated our efforts on this population.

- Figure 3: It would be important to also analyze bone marrow in the in vivo adoptive transfer setting to see if ILCPs could also relocate to the bone marrow.

We have analysed the progeny found in the bone marrow of ILCP recipients and include data in a new figure on the phenotype of the group 1 ILCs produced (Figure 7). We found no evidence of donor ILCPs in the bone marrow but since this would represent a tiny

proportion of any ILC population and the number of donor cells is small we cannot exclude that a handful are present but are missed due to sampling error. Thus, whilst it is very challenging to prove a negative, we have no evidence for the presence of these cells in the BM.

Furthermore, inclusion of a control population would help to identify positive and negative populations in the flow cytometric representatives (NKp46 vs NK1.1 and Eomes vs NK1.1 plots).

We have now included additional supplementary figures to demonstrate how positive and negative staining for NKp46, NK1.1 and Eomes were defined (Figure S5b-c).

Besides mature ILC lineages, can ILCPs still be detected?

In our analysis of the ILCP progeny we found no evidence of donor ILCPs in the tissues analysed.

- Figure 5: a, the dimensionality reduction (UMAP) should be indicated in the plot. In addition, it would be relevant to set mature ILC-related markers (Robinette et al., PMID: 25621825; Gury-BenAri et al., PMID: 27545347) into context.

We thank the reviewer for alerting us to this omission, this has now been corrected. We have also added more information to the original figure and added another figure to put the gene expression pattern of our progeny clusters in context with respect to the studies of Robinette et al. and Gury-BenAri et al. (Figure 5a and b, Lines 236-253; Figure S7, Lines 251-253; Table S2).

- Figure 6: How are lineage determining transcription factors such as GATA3 and T-bet expressed throughout the scRNAseq dataset?

We have now included an additional figure to show this data and these markers are also included in our high dimensional spectral cytometry antibody panel (Figure S10 Lines 315-319: Figure 9).

A) is lacking the unit in the color legend

This has been corrected in Figure 5b.

- Figure S1: ILCs, especially ILC3, are described to express lower levels of CD45 (e.g., Zhou et al. PMID: 30944470 Ext. Data 3). Would inclusion of cells expressing CD45 intermediately increase the ILC(P) populations? Following the gating strategy, the lineage gating is not very convincing as it is hard to evaluate positive and negative cells, maybe plotting lineage vs. Id2-BFP or IL-7Ra would help to identify a negative lineage. Similarly, the KLRG1-CCR6-NKp46- gate is unclear, maybe plotting APC/EF660 vs RORgt-Kat would help to identify a negative population. In particular, gating of Bcl11b-Tomato vs PLZF-Citrine should be revised and question the purity of sorted populations for the in vitro assays in Figure 1c-e.

We have now included extra components in our supplementary figure demonstrating our sorting strategy to address these queries and justify the definition of our gates (Figure S1). The appearance of a CD45 "low" population was an artefact of the axis and has been rectified. We thank the reviewer for the suggestion of plotting CCR6/KLRG1/NKp46 against Rorgt-Katushka. In our extended figure we demonstrate that if we do this the double negative population is the same as that defined by our original gating strategy. However, we did recognise that this was a more elegant approach and have utilised it in the additional experiments.

- Figure S3: Can ILCPs based on Plzf and Bcl11b also be identified in other organs analyzed (gating adjusted to tissue-specific ILC markers)?

We have now done a survey of a number of tissues for cells that fall into our ILCP gate and phenotyped them using high dimensional spectral analysis and include these data in Figure 9 and Figure S12 which are referred to in Lines 344-367.

- Figure S6: It is very hard to draw any conclusion from the data as they are presented. A heatmap showing the gene expression pattern per cluster would help to see differences and overlapping genes between siILCP and progeny. Expression of selected genes as violin plot would help the reader to evaluate overall expression within the individual clusters.

We thank the reviewer for their suggestion and we have now re-analysed the data and present it in 2 new figures both as an average per cluster and on a cell-by-cell basis. We have also adapted our violin plot figure of key ILCP related genes to separate out the expression in the progenitors and each progeny cluster. Please see Figure 5c and Lines 310-313; Figure S7 and Lines 313-315; Figure 8a and Lines 321-338).

- Figure legends: n should always be stated

We believe we have remedied any omissions.

- Discussion: "Although we identified expression of the IL-25 receptor by some siLP-ILCPs, in vitro inclusion of IL-25 in siLP-ILCPs cultures, which is produced by the gut epithelium (but less so in the lung) and induces ILC2 expansion, did not alter their differentiation." Please provide a reference for IL-25 levels in lungs vs intestine.

We thank the reviewers for highlighting our lack of clarity. However, we have now decided that in the light of the additional data we now include in the manuscript this data does not add significantly to the key message of the paper and we have removed it.

- Several important citations about human and murine tissue ILCP are missing. Please revisit the bibliography and include at least:

These references are now included in the introduction and where appropriate in the discussion.

-Chea et al., PMID: 26832410: mouse fetal spleen and mLN ILCP

Ref. 22

Lines 84, 366 and 398

-Ghaedi et al., PMID: 3181-6636: murine adult and neonatal lung

Ref. 28

Lines 94, 215, 336, 359 and 402

-Liu et al., PMID: 34239074: human fetal tissues from 8, 10, and 12 weeks PCW, specifically the liver, thymus, spleen, intestine, skin, and lung

Ref. 16

Lines 71 and 411

-Montaldo et al., PMID: 25500367, human adult tonsil CD34+ RORgt+ progenitors

Ref. 20

Line 80

-Scoville et al., PMID: 27178467, human secondary lymphoid tissue CD34+ RORgt+ ILCP

Ref. 21

Line 80 and 411

-Simic et al., PMID: 32783932: mouse fetal lymph node ILCP

Ref. 25

Line 88, 366 and 398

-Stehle et al., PMID: 34556887: mouse fetal intestine ILCP

Ref. 24

Line 87

-Suo et al., PMID: 35549310, human prenatal intestine scRNA ILCP

Ref. 17

Line 72

Reviewer #3 (Remarks to the Author):

In this manuscript, Clark et al. employed a multicolor fluorescent reporter mouse model, for Id2, PLZF, Bcl11b and Rorgt, to identify ILC progenitors residing in the small intestine. Next, the ability of these precursor cells to generate distinct NK/ILC lineages was assessed using in vitro and in vivo systems. The authors found a siLP-ILCP population able to give rise to a distinct NK/ILC lineages in the small intestine, while the potential to generate NK/ILCs in other tissues, namely in spleen, liver and lung was limited to Eomes+ cells.

The manuscript is interesting and helps fuel the current view of ILC-poiesis in the tissues. Some issues emerged that need to be clarified (discussed below).

1. A small fraction of siLP Gata3^{bright} ILC2 does not express KLRG1. Is it possible that some of the lin⁻ Id2⁺IL7R⁺NKp46⁻NK1.1⁻KLRG1⁻Rorgt⁻CCR6⁻ cells shown in Figure 1b contain these KLRG1-Gata3^{bright} ILC2? This might explain the enrichment of Bcl11b⁻Tomato⁺ cells in the contour plot, or the results presented in Figure 4.

We thank the reviewer for their comment. It is possible that some of the P+B⁺ and P-B⁺ cells are KLRG1 negative ILC2s and consequently we did not analyse these cells further. The output shown in Figure 4 is only of the P+B⁻ cells and we now include and additional data that demonstrate that these cells express Gata3 only at low levels (Figures S10, Lines 315-319; and Figure 9).

2. Are siLP-ILCP found in the large intestine?

We have now done a survey of a number of tissues for cells that fall into our ILCP gate and phenotyped them using high dimensional spectral analysis and include these data in the results. We found no evidence of cells with our phenotype in the colonic LP (Figure S12).

3. Related to Figure 3A. siLP-ILCP population seems to give rise mainly Eomes⁺ NK cells in spleen, lung and liver. Do siLP-ILCP lose the capacity to generate Eomes⁻ ILC1 in vivo?

We thank the reviewer for pointing out a lack of clarity on our part. Whilst the majority of progeny in the liver, lung and spleen are NK1.1⁺ Eomes⁺ they are not exclusively so and in the siLP there are also some Eomes⁻ NK1.1⁺ cells. We have added data to our supplementary figure to show the percentages of Eomes⁻ NK1.1⁺ cells across tissues (Figure S6, Lines 181-185).

4. Moreover, the small intestine is generally enriched in Eomes⁻ ILC1. Can siLP-ILCP generate Eomes⁻ ILC1 in this tissue? Few more markers helping characterize the phenotype of the ILC1/NK cells generated in vivo from the siLP-ILCP are needed (CD49a, CD49b, Ly49s and others), at least in one representative tissue.

The presence of Eomes⁻ NK1.1⁺ progeny in the siLP is as described in the previous response. We have now included additional analysis of our scRNAseq data to determine the pattern of expression of the suggested markers (Figure S8c, Lines 266-271). Moreover, we now include a new results section (Figure 7, Lines 279-298) with a detailed multi-parameter (including the suggested markers and others) spectral analysis of the type 1 progeny across multiple recipient tissues.

5. Figure S6. To evaluate signs of early differentiation in siLP-ILCP, the authors should use gene signatures from an independent dataset (Immgen?). The current analysis is quite circular and the authors might lose important information. In alternative, siLP-ILCP should be excluded when defining ILC-specific signatures.

We thank the reviewer for this suggestion and we have now re-analysed our scRNAseq data for the progenitors in the context both of published mature ILC gene expression signatures and our own most differentially expressed genes for our progeny. This analysis is now included in two new figures and more clearly demonstrates that there is no evidence of sub-

clustering of our progenitors that suggest embarkment upon a committed trajectory (Figure 5c, Lines 310-313; Figure S7, Lines 313-315).

6. The scRNA-seq data could be further analyzed to be more informative. Is there any combination of surface markers/molecules helping to gate siLP-ILCP without using reporter mice?

We have now included gene expression analysis identifying genes that are most highly differentially expressed between the progenitors and their progeny to address this query. Some of these genes do encode cell surface molecules. However, without the use of our transcription factor reporters we believe it would be difficult to distinguish the siLP-ILCPs from the any other haemaopoietic cells found in the siLP (Figure 8b and c, Lines 339-442).

Is it possible to define the signature of siLP-ILCP and look for these cells in human gut datasets? These infos would be precious for further studies.

We have now included gene expression analysis identifying genes that are most highly differentially expressed between the progenitors and their progeny to address this query (Figure 8b and c, Lines 339-442). However, our analysis across different tissues in the mouse would suggest that generating a more detailed signature that would hold true in human gut tissues would be challenging and is beyond the scope of this paper. The scRNAseq data will be available on public databases to allow others to perform such analyses (Figure 9, Lines 344-367 and Lines 407-413).

Minor

- Page 4; lanes 99-101. Citation or FACS plot should be added at the end of this sentence.

We now include a citation on line 115.

- Figure 2b. I feel that keeping lin⁺ cells in the main Figure is confusing. Moreover, I feel the authors should choose just one way to represent the flow data (contour plot?) in the same figure, for consistency.

We have amended the figure as requested and moved the data for the lineage positive cells to supplementary data (Figure 2a and Figure S4b).

- Information regarding scRNA-seq analysis is very limited.

We have added more information to the methods section relating to the scRNAseq analysis (Lines 603-619).

REVIEWER COMMENTS

Reviewer #1 (Remarks to the Author):

The authors have extensively revised the manuscript and provide important novel analyses. All my major concerns have been addressed. The new data are interesting and further improve the manuscript. I have minor comments and suggestions to provide context to the new analyses and to improve their representation:

- There are two new papers in by the Bhandoola and Bendelac groups in Nature immunology dissecting ILC1 and NK cell precursors and heterogeneity. Authors should include a short discussion how these newly identified subsets relate to the siLP-ILCP identified here, and to the ILC1/NK progeny data presented in their manuscript.

- Discussion of ILC-NK spectrum, lines 255-277: CD49a is broadly regulated and is expressed by tissue-resident NK cells in salivary glands or tumors, in combination with ly49 receptors. These cells can also express Kit and IL21r. To me, Fig S9 look like having circulating NKs in lung, and resident NK and ILC1 in SILP. Maybe this possibility could be considered in their current interpretation and discussion of these data

Figure 7:

7c- this heat map seems to plot MFIs, are these a function of increased/decreased % of cells expressing respective markers, or of higher protein levels on all analyzed ILCP subsets? Could you also plot % of expression of the respective markers? This could be two separate heat-maps, or, alternatively, a dot plot format could probably integrate both, % and MFI?

Could the low level of detection of some Ly49 receptors reflect cleavage during digestion?

Figure 9:

Thank you for providing this analysis, I think this is a valuable resource for the community. Would it be possible to provide an estimate of detected numbers of ILCPs (SI-like ILCP, not the subclusters) per analyzed organ?

9b: as in 7c - this heat map seems to plot MFIs, are these a function of increased/decreased % of cells expressing respective markers, or of higher protein levels on all analyzed ILCP subsets? Could you also plot % of expression of the respective markers? This could be two separate heat-maps, or, alternatively, a dot plot format could probably integrate both, % and MFI?

9c What do the individual UMAPs depict? The colour codes from tissues are difficult to distinguish on the UMAPs (this works much better in 7c). Could you plot one UMAP per tissue and highlight where cells from the respective tissue locate on the UMAP (I am not sure if this is what is shown)?

Figure S12:

There seems an error in the gating strategy in S12a: CD45+ cells are gated as Zombie NIR – cells, then the next plot (gating versus CD11b) again contains dead cells.

S12b: what is the parental gate? Can authors provide an estimate of detected numbers of the ILCPs per analyzed organ?

Reviewer #2 (Remarks to the Author):

This is a revised version of the manuscript of Clark et al entitled "Intestinal ILC progenitors differentially generate ILC subsets dependent on the recipient tissue microenvironment". The authors have thoroughly addressed the main criticisms raised, and I believe the paper provides important novel insights in the field.

Reviewer #3 (Remarks to the Author):

The authors have addressed all my concerns.

REVIEWER COMMENTS

Reviewer #1 (Remarks to the Author): The authors have extensively revised the manuscript and provide important novel analyses. All my major concerns have been addressed. The new data are interesting and further improve the manuscript. I have minor comments and suggestions to provide context to the new analyses and to improve their representation:

- There are two new papers in by the Bhandoola and Bendelac groups in Nature immunology dissecting ILC1 and NK cell precursors and heterogeneity. Authors should include a short discussion how these newly identified subsets relate to the siLP-ILCP identified here, and to the ILC1/NK progeny data presented in their manuscript.

We have now inserted the following text into the Discussion (lines 447-458) to address these papers:

“Recent studies have identified bone marrow NK cell progenitors that develop independently of PLZF⁺ ILCPs^{56,57}. Both studies highlight differences between the NK cells derived from their novel NKPs and BM-ILCPs^{56,57}. Interestingly, like siLP-ILCPs, other than in the intestine, the recently reported progenitors are biased towards the production of NK cells *in vitro* and *in vivo*. However, since the ENKPs described by Ding *et al.* are Flt3⁺ and PLZF⁻⁵⁷, and the cells described by Liang *et al.* are Eomes^{hi}NK^{neg}⁵⁶, these populations are phenotypically distinct from siLP-ILCPs. Furthermore, whilst the characterised ILCP_NK cells have similarities with the group 1 ILC progeny of the siLP-ILCPs, a number of the genes used to discriminate ENKP_NK cells from ILCP_NK cells are common to the genes which distinguish the siLP-ILCP lung from the siLP progeny⁵⁷. Thus, in the context of our data these studies emphasise the complexities and nuances of group 1 ILC development which is influenced by both progenitor phenotype and tissue location.”

- Discussion of ILC-NK spectrum, lines 255-277: CD49a is broadly regulated and is expressed by tissue-resident NK cells in salivary glands or tumors, in combination with ly49 receptors. These cells can also express Kit and Il21r. To me, Fig S9 look like having circulating NKs in lung, and resident NK and ILC1 in SILP. Maybe this possibility could be considered in their current interpretation and discussion of these data

We thank the reviewer for proposing this alternative interpretation. We have now included reference to this possibility in the section mentioned (lines 262-263 and 275-281) along with additional relevant citations (47 - 49).

Figure 7:

7c- this heat map seems to plot MFIs, are these a function of increased/decreased % of cells expressing respective markers, or of higher protein levels on all analyzed ILCP subsets? Could you also plot % of expression of the respective markers? This could be two separate heat-maps, or, alternatively, a dot plot format could probably integrate both, % and MFI?

The heatmap in Figure 7b depicts MFI of each cluster for the indicated molecules. Since this heatmap is a result of the high dimensional analysis of the clusters in an unbiased manner, the relative MFI depicted is a function of both proportion of cells expressing each marker and the protein level of the positive cells within said cluster. Each marker would behave differently (having higher or lower % of positive cells +/- higher or lower expression of the protein in positive cells). To help with the interpretation of the data, we have now included a heatmap depicting the % of expression of the respective markers in the additional Supplementary Figure 9a. We have also indicated in the Figure 7b that scaled MFI is depicted in these heatmaps and included additional clarification in the figure legend (lines 765-766). We believe that with so many markers and in some cases small populations use of two-dimensional dot plots would be unhelpful.

Could the low level of detection of some Ly49 receptors reflect cleavage during digestion?

We cannot absolutely exclude that the digestion steps required to generate single cell suspensions of lung and siLP cells may impact the apparent cell surface expression of some Ly49 receptors. However, whilst the collagenase used may have low levels of other proteases none of the Ly49 receptors investigated have collagenase enzyme substrate sites. Further, Ly49C/I expression is low across all clusters represented in all tissues including those which do not require digestion (BM, liver and spleen). Moreover, we also found in the scRNAseq data from recipient siLP and lung that the transcript for Ly49D (*Klra4*), which is low across all clusters except the smaller outlier cluster LP5, is hardly expressed at all and the transcript for Ly49H (*Klra8*), which is low across the majority of clusters, is also at low levels. These expression data would seem to confirm the flow data. We now include these additional scRNAseq data in supplementary figure 8c (referred to in line 271).

Additionally, we present here flow cytometric analysis of the expression of Ly49 receptors by group 1 ILCs in the lung and siLP of the parental 4 colour reporter mice which were processed and analysed in parallel with the siLP-ILCP recipient mice and demonstrate that these Ly49 receptors can be detected even after tissue processing involving enzymatic digestion.

Figure 9:

Thank you for providing this analysis, I think this is a valuable resource for the community. Would it be possible to provide an estimate of detected numbers of ILCPs (SI-like ILCP, not the subclusters) per analyzed organ?

We have now calculated an approximate average number of cells with the ILCP-consistent phenotype across the 6 animals analysed for those tissues where they were found, except for bone marrow where not all bones were sampled and it would therefore be difficult to calculate this value. This information is now included in Supplementary Figure 12c (referred to in lines 355-357).

9b: as in 7c - this heat map seems to plot MFIs, are these a function of increased/decreased % of cells expressing respective markers, or of higher protein levels on all analyzed ILCP subsets? Could you also plot % of expression of the respective markers? This could be two separate heatmaps, or, alternatively, a dot plot format could probably integrate both, % and MFI?

As for Figure 7c the heatmap in Figure 9b depicts MFI of each cluster for the indicated molecules. Please see above for the details. To help with the interpretation of the data, we have now included a heatmap depicting the % of expression of the respective markers in the additional Supplementary Figure 12d. We have also indicated in the Figure 9b that scaled MFI is depicted in these heatmaps and included additional clarification in the figure legend (lines 788-789). We believe that with so many markers and in some cases small populations use of two-dimensional dot plots would be unhelpful.

9c What do the individual UMAPs depict?

These UMAPS depict the range of phenotypes of the $CD45^+lineage^-Id2-BFP^+IL-7Ra^+NK1.1^-NKp46-KLRG1^-Rorgt-Katushka^-CCR6-PLZF^+Bcl11b^-$ cells from all the tissues where they were

found, and then combined in order to allow for more meaningful clustering, especially as some tissues have very few of these cells. In Figure 9c the first UMAP combines cells from all tissues and we then separate the cells from each tissue, each with a different colour, and map them onto the this UMAP (in grey in the background) to give a representation of their distribution across the UMAP. This information is now included in the figure legend (lines 790-793). The contribution of each cluster to the cells from each tissue is then represented in the heatmap in Figure 9d. We apologise for the lack of a colour key for this image which has now been corrected.

The colour codes from tissues are difficult to distinguish on the UMAPs (this works much better in 7c).

We recognise that due to the small number of cells from some tissues it is not easy to see this (Figure 7c had many more cells from each tissue). We have now moved the colour key for the tissues to act as a label for each UMAP and have endeavoured to improve the appearance of the coloured dots to make them clearer.

Could you plot one UMAP per tissue and highlight where cells from the respective tissue locate on the UMAP (I am not sure if this is what is shown)?

We believe if we attempted to cluster the cells on a tissue-by-tissue basis there would be too few cells to be instructive, if this is what the reviewer is suggesting. Otherwise, this is what we have shown and for clarity we have expanded the explanation in the figure legend (lines 790-793).

Figure S12: There seems an error in the gating strategy in S12a: CD45⁺ cells are gated as Zombie NIR – cells, then the next plot (gating versus CD11b) again contains dead cells.

We thank the reviewer for alerting us to this error. The y axis in this plot should have been labelled Lineage-AF700. This error has now been corrected on the Supplementary Figure 12a and lineage is defined in the figure legend (line 947).

S12b: what is the parental gate?

We thank the reviewer for alerting us to this ambiguity. The parental gate is CD45⁺lineage⁻Id2-BFP⁺IL-7Ra⁺NK1.1⁻NKp46⁻KLRG1⁻Rorgt-Katushka⁻CCR6⁻ as defined in the gating strategy shown in Supplementary Figure 12a. This information is now explicitly given in the figure legend (line 951-952)

Can authors provide an estimate of detected numbers of the ILCPs per analyzed organ?

We have now calculated an approximate average number cells with the ILCP consistent phenotype for those tissues where they were found across the 6 animals analysed, with the exception of bone marrow where not all bones were sampled and it would therefore be difficult to calculate this value. This information is now included in Supplementary Figure 12c (referred to in lines 355-357).

REVIEWERS' COMMENTS

Reviewer #1 (Remarks to the Author):

Thank you for clarifying the remaining points, and congratulations—I believe the paper offers significant novelty and importance!

Response to reviewers of Clark et al NCOMMS-23-25926B

We believe we have dealt with all comments from reviewers 2 and 3 in previous resubmissions.

We thank reviewer 1 for their kind words and for their valuable input to the revisions of our paper.